# Thermodynamic and dynamic responses of the hydrological cycle to solar dimming

Jane E. Smyth[1], Rick D. Russotto[2], and Trude Storelvmo[1]

[1]Department of Geology & Geophysics, Yale University, New Haven, Connecticut, USA
[2]Department of Atmospheric Sciences, University of Washington, Seattle, Washington, USA

*Correspondence to:* Jane E. Smyth (jsmyth@princeton.edu)

**Abstract.** The fundamental role of the hydrological cycle in the global climate system motivates thorough evaluation of its responses to climate change and mitigation. The Geoengineering Model Intercomparison Project (GeoMIP) is a coordinated international effort to assess the climate impacts of solar geoengineering, a proposal to counteract global warming with a reduction of incoming solar radiation. We assess the mechanisms underlying the rainfall response to a simplified simulation of such solar dimming (G1) in the suite of GeoMIP models and identify robust features. While solar geoengineering nearly restores preindustrial temperatures, the global hydrology is altered. Tropical precipitation changes dominate the response across the model suite, and these are driven primarily by shifts of the Hadley circulation cells. We report a damping of the seasonal migration of the intertropical convergence zone (ITCZ) in G1, associated with preferential cooling of the summer hemisphere, and annual mean ITCZ shifts in some models that are correlated with warming of one hemisphere relative to the other. Dynamical changes better explain the varying tropical rainfall anomalies between models than do changes in relative humidity or the Clausius-Clapeyron scaling of precipitation minus evaporation ($P - E$), given that the relative humidity and temperature responses are robust across the suite. Strong reductions in relative humidity over vegetated land regions are likely related to the $CO_2$ physiological response in plants. The uncertainty in the spatial distribution of tropical $P - E$ changes highlights the need for cautious consideration and continued study before any implementation of solar geoengineering.

## 1 Introduction

Solar geoengineering has been proposed as a way to counter the effects of global warming induced by anthropogenic greenhouse gas emissions (e.g., Crutzen, 2006; Robock et al., 2009). By reducing incoming solar radiation, solar geoengineering would bring the climate with elevated concentrations of $CO_2$ into radiative balance. It would compensate for a change in surface temperature from increased $CO_2$ trapping of outgoing longwave radiation with a reduction of incoming shortwave radiation. Solar geoengineering is a controversial proposal, but should it come into favor due to continued greenhouse gas emissions, it is critical that the climate effects be understood before deployment (NRC, 2015).

The Geoengineering Model Intercomparison Project (GeoMIP) is intended to determine robust responses of the climate to various simulations of solar geoengineering, in experiments that range from simple representations of the solar dimming, to realistic representations of stratospheric aerosol emissions or marine cloud brightening (Kravitz et al., 2010). The GeoMIP

experiments are based on the Coupled Model Intercomparison Project Phase Five (CMIP5), which is a protocol for experiments using coupled atmosphere-ocean climate models (Table 1). The GeoMIP G1 experiment counteracts the forcing from quadrupled atmospheric $CO_2$ levels with a simple reduction of the solar constant across all wavelengths. The G1 experiment was run from the steady state preindustrial control (piControl) run, followed by an abrupt quadrupling of $CO_2$, and a simultaneous solar constant reduction for 50 years. The idealized nature of this simulation is conducive to multimodel comparison. It superimposes two large and opposite climate forcings, which offset one another nearly completely in terms of global mean net radiation balance at the top of the atmosphere and near-surface atmospheric temperature, but do not cancel in their hydrological effects, especially on local scales (Kravitz et al., 2013b).

We analyze twelve fully coupled models from the G1 experiment (Table 1). There are serious errors in the precipitation output files from the EC-Earth model and it is thus excluded from any analysis involving the precipitation field. The models differ in their ocean, ice sheet, land surface and atmospheric components. The latter two components are particularly relevant for this study. Some, but not all models, feature dynamic vegetation distributions. The 11 models include a wide range of parametrizations and configurations, allowing for strong conclusions about robust climate responses that appear across models (Kravitz et al., 2013a).

The water cycle impacts agriculture, economies, as well as the welfare of ecosystems and human civilizations (IPCC, 2014). It is imperative to understand the effects of solar geoengineering on global hydrology, to evaluate whether the risks or unintended consequences of such an approach are likely to outweigh the benefits.

While Bala et al. (2008), Schmidt et al. (2012), and Kravitz et al. (2013b) have reported the uncertainty of tropical rainfall responses to geoengineering, no previous authors have presented the Hadley circulation changes in the annual mean or seasonally. Davis et al. (2016) found that the Hadley cell edge latitudes do not change in G1 relative to piControl, but do not examine changes in fluid motions within the Hadley cell. Bala et al. (2008) evaluate a single model and discuss how tropical precipitation might be suppressed when insolation is reduced because this cools the surface relative to the overlying atmosphere, stabilizing the troposphere and reducing convection. Insolation changes affect the surface energy budget more than greenhouse gases, and thus necessitate a stronger response by the sum of latent and sensible heat fluxes. Schmidt et al. (2012) examine ITCZ changes in G1, but only in four of the GeoMIP models, and they do not attempt to explain causes of variability within the suite. They find that the global average precipitation increase in abrupt4xCO2 is about two times larger than the precipitation reduction in G1, but note that the precipitation change in G1 is still substantial. By assessing changes in the surface and atmospheric energy budgets, Kravitz et al. (2013b) conclude that precipitation changes are mostly explained by evaporation changes, implying that annual mean circulation changes are likely small. They identify an analysis of circulation changes in G1 as a fruitful future research direction. We build upon their findings by analyzing the Hadley circulation changes in G1 on both annual mean and seasonal timescales.

Kleidon et al. (2015) also underscore the importance of the surface energy balance in making robust predictions of the hydrological effects of radiative forcing. They decompose the hydrological response into fast and slow components, and infer hydrologic changes using analytic expressions of physical constraints. Our study, on the other hand, focuses on the steady state response and utilizes decomposition to understand simultaneous physical response modes. Tilmes et al. (2013) note reduced

global evaporation in the G1 ensemble, and a reduction in global precipitation of approximately 4.5%, with stronger reductions in monsoon regions. Precipitation extremes are reduced by around 20% in G1.

This paper makes progress towards understanding the global impacts of geoengineering by analyzing the thermodynamic, relative humidity, and dynamic components of the hydrological change. We identity robust conclusions across the suite, and present a possible explanation for the discrepancies in tropical rainfall shifts. We assess the contributions of several different effects to changes in precipitation minus evaporation ($P - E$) in the GeoMIP G1 experiment, as follows:

1. In Section 2.1, we analyze the thermodynamic response of $P - E$ to geoengineering.

2. In Section 2.2, we assess the role of changes in relative humidity on $P - E$.

3. In Section 2.3, we investigate the extent to which atmospheric circulation patterns, namely changes in the Hadley cell strength and position, drive $P - E$ changes in the models on both annual and seasonal timescales.

## 2   Analysis & Results

### 2.1   Thermodynamic Scaling of $P - E$

Precipitation minus evaporation determines the soil moisture and the amount of runoff on land, and is crucial in setting the salinity of the mixed layer of the ocean (Byrne and O'Gorman, 2015). We here discuss the component of $P - E$ changes driven by residual surface temperature changes (G1- piControl). Surface heating increases the temperature and the evaporation rate, which increases the atmospheric moisture content, or specific humidity $q$ (e.g., Trenberth, 1999). We have confidence about certain aspects of the hydrological cycle's response to greenhouse gas warming, particularly those tightly coupled to the increase in saturation vapor pressure with warming (Held and Soden, 2006). The Clausius-Clapeyron expression (Eq. (1)), where $R$ is the gas constant, $L$ is the latent heat of vaporization, and $\alpha$ is the Clausius-Clapeyron scaling factor, relates the derivative of the natural log of saturation vapor pressure $e_s$ with respect to temperature ($T$) to temperature itself.

$$\frac{d\ln e_s}{dT} = \frac{L}{RT^2} \equiv \alpha(T) \tag{1}$$

At typical near-surface temperatures, saturation vapor pressure increases at $7\ \%\mathrm{K}^{-1}$.

Precipitation minus evaporation follows Clausius-Clapeyron scaling, as in Eq. (2), where $\delta$ indicates the change between climate states, given three important assumptions (Held and Soden, 2006).

$$\delta(P - E) = \alpha\,\delta T\,(P - E) \tag{2}$$

First, it assumes small meridional and zonal gradients of temperature anomalies relative to $P - E$. Second, the relationship assumes that there is no change in near-surface relative humidity between climate states, and that the total moisture flux divergence in the atmosphere scales with near-surface specific humidity. Third, it assumes that there is no change in the atmospheric flow. Though it is known that relative humidity and atmospheric circulation are not constant in a changing climate, the thermodynamic scaling is a useful way to represent the role of a simple physical mechanism (i.e., the Clausius-Clapeyron scaling

of saturation vapor pressure with temperature) on global $P - E$ anomalies (Byrne and O'Gorman, 2015). This thermodynamic scaling equation represents the component of $P - E$ change driven directly by surface temperature perturbations.

This study evaluates the extent to which the basic physical relation between saturation vapor pressure and temperature accounts for the hydrological response to a combination of large-magnitude forcings: greenhouse gas warming and solar dimming.

We investigate how well thermodynamic scaling predicts hydrologic changes in a geoengineered climate for each model by comparing the prediction using Eq. (2), calculated in each grid box with annual mean data and then averaged zonally, to the annual and zonal mean $P - E$ anomaly between G1 (years 11-50) and piControl (all years) in the model simulations. We also consider the annual-mean global distribution of precipitation minus evaporation anomalies.

To provide reference points for our analysis, we have re-plotted some thermodynamic variables in Figures 1-3 that originally appeared in the G1 overview paper by Kravitz et al. (2013a). The experimental design results in small temperature anomalies (relative to climatological temperatures) between G1 (years 11-50) and piControl (all years) (Fig. 1), with less than 1 K of residual temperature change across most of the globe. The ensemble mean change in $P - E$ shows greater hydrological changes (up to 1 mm/day) in the tropics than at higher latitudes (Fig. 2). Figure 3, which separates the precipitation and evaporation changes, reveals that most of the spatial structure in the $P - E$ anomaly comes from the precipitation change.

In contrast, the thermodynamic scaling captures virtually no change in global $P - E$ patterns, since by experimental design the temperature anomaly is small between the G1 and piControl scenarios (Fig. 4B). The ensemble mean temperature anomalies between G1 and piControl show residual warming exceeding 1 K at high latitudes and cooling at low latitudes as a robust feature across the suite (Fig. 1) (Kravitz et al. 2013a). Such temperature anomalies are generally not sufficient to generate appreciable thermodynamic changes in $P - E$. Deviations of the ensemble mean simulated precipitation minus evaporation anomaly from the thermodynamic scaling are largest in the tropics, where temperature anomalies are small (Fig. 4). In BNU-ESM, thermodynamic scaling predicts a $P - E$ enhancement over the anomalously warm high latitudes, where the temperature response to quadrupled $CO_2$ levels is poorly compensated by solar dimming (annual mean G1-piControl anomalies >2K at polar latitudes, results not shown here).

The ensemble mean precipitation response reflects strong reductions in subtropical precipitation across the Pacific Ocean (exceeding 0.8 mm/day) (Fig. 3). The precipitation changes are larger in magnitude than the evaporation changes, so this spatial structure is apparent in the ensemble mean $P - E$ as well, with drying around 15°N and 15°S across the Pacific Ocean (Fig. 2). Previous research has suggested that this is a result of the nature of the G1 experiment forcing. Solar geoengineering might suppress tropical precipitation since the reduction in shortwave radiation cools the surface more than the mid-troposphere, increasing atmospheric stability and reducing convection (Bala et al., 2008). However, individual model behavior is not consistent with this ensemble mean picture of suppressed off-equatorial precipitation. Rather, the zonal mean $P - E$ shifts in different directions in individual models so that higher amplitude changes cancel out in the ensemble mean (Fig. 4A). The HadCM3, HadGEM2-ES, and CESM-Cam5.1-FV models show $P - E$ anomalies indicating a northward shift in the Intertropical Convergence Zone (ITCZ), while those of GISS-E2-R, Can-ESM2, and MIROC-ESM demonstrate a southward shift. Annual mean anomalies in the zonal mean $P - E$ exceed 0.6 mm/day in the GISS-E2-R and HadGEM2-ES simulations. In CCSM4, IPSL-

CM5A-LR, and NorESM1-M models, the ITCZ appears to narrow, with precipitation increasing at the equator and decreasing near 10°N and 10°S.

To better understand the role of relative humidity ($H_s$) changes in the hydrological response to G1, we investigate the contribution of local changes in $H_s$ to $\delta(P-E)$ as well as the global distribution of annual mean $H_s$ changes in the following

section. We will then investigate the dynamical changes in the tropics in Section 2.3.

## 2.2    Relative Humidity

The simple thermodynamic scaling described above (Eq. 2) assumes no changes in relative humidity between climate states. In this section, we assess the role that relative humidity changes play in the $P-E$ response to uniform solar dimming. Relative humidity is the ratio of actual vapor pressure to saturation vapor pressure ($\frac{e}{e_s}$), or almost equivalently, specific humidity to

saturation specific humidity ($\frac{q}{q_s}$). It can change with the water availability or temperature, with the latter affecting the saturation vapor pressure as in Eq. (1). The atmospheric boundary layer provides moisture to the free troposphere, where water vapor plays an important role in radiative transfer, the hydrological cycle, and climate sensitivity (Willett et al., 2010). The near-surface relative humidity parameter is also of interest in climate change studies for evaluating the risk of human heat stress, under both high and low $H_s$ extremes (Sherwood et al., 2010; Souch and Grimmond, 2004).

The assumption of constant relative humidity in the simple thermodynamic scaling of $P-E$ (Eq. (2)) relies on the availability of moisture. In a moisture-limited regime (i.e., over land) $q$ may not increase proportionally with temperature, breaking the assumption of constant relative humidity. Under this circumstance, relative humidity adjustments would contribute to changes in the $P-E$ between climate states. An observational study found decreasing surface relative humidity from 1998-2008 over low and midlatitude land areas due to inhomogeneities in surface heating and moisture availability (Simmons et al., 2010).

This was corroborated by a later observational study, though the global long-term relative humidity trend was statistically insignificant (Willett et al., 2014). Previous studies have proposed that simulated and observed land-sea contrasts in relative humidity responses to global warming can be explained by the stronger temperature-driven increase in saturation specific humidity over land, which is not sufficiently compensated by moisture transport from the ocean (Byrne and O'Gorman, 2016). Byrne and O'Gorman (2016) develop a conceptual box model which quantitatively supports that this ocean-control mechanism,

as well as changes in evapotranspiration, explain simulated relative humidity anomalies over land.

To better understand the contribution of local relative humidity changes to the $P-E$ response, we calculated an "extended scaling" adapted from Byrne and O'Gorman (2015). Our extended scaling includes the first two terms from Byrne and O'Gorman's equation,

$$\delta(P-E) = \alpha\,\delta T\,(P-E) + \frac{\delta H_s}{H_s}\,(P-E) \tag{3}$$

where $H_s$ is the relative humidity at the surface. The calculation takes local changes in $H_s$ into account, but it excludes the horizontal gradients of changes in $H_s$ and $T$ since the moisture flux calculation requires daily mean model output, which was not archived for the G1 experiment for most models. We calculated the difference between the zonal mean $P-E$ anomalies in

the extended and simple scalings to quantify the influence of local changes in $H_s$. We also calculated the difference between simulated $P - E$ anomalies and the extended thermodynamic scaling. Relative humidity data for this analysis were unavailable for CESM, HadC, and MPI due to limited functionality of the central GeoMIP model data server, the Earth System Grid Federation (ESGF).

The deviations of the extended scaling from the simple scaling are less than $0.1$ $\mathrm{mm/day}$ in all models (Fig. 4C). This demonstrates that the local changes in relative humidity under solar dimming (ignoring the gradient of $H_s$ changes) play at most a modest role in the zonal mean $P - E$ response. Local relative humidity changes were also found to be of minimal importance to $P - E$ anomalies in global warming simulations (Byrne and O'Gorman, 2015). Figure 4D indicates that most of the zonal mean $P - E$ anomalies are not captured by the Clausius-Clapeyron scaling or by local relative humidity changes.

We therefore attribute the residual simulated $\delta(P - E)$ to atmospheric circulation changes, or gradients of changes in $H_s$ and $T$. Despite the limited impact of local relative humidity changes on zonal mean $P - E$ changes, regional impacts could still be large and important.

  To supplement this analysis, we consider the absolute changes in the relative humidity distribution to explain $P - E$ anomalies between G1 (years 11-50) and piControl (all years) simulations unaccounted for by thermodynamic or dynamic mecha-

15 nisms. In six of the eight models presented here, relative humidity is reduced over land and conserved over ocean (Fig. 5). The relative humidity reductions are largest over tropical South America and sub-Saharan Africa. The reductions are up to $15\%$ (0.15) in GISS-E2-R and HadGEM2-ES (calculated as the G1 relative humidity ($\%$) minus the piControl relative humidity ($\%$)). Relative humidity is not uniformly reduced over land. Over the deserts of Saudi Arabia, northern Africa, and Australia, relative humidity changes are negligible or in some models, slightly positive. A similar spatial pattern is evident in the evap-

20 oration anomaly field, with the most strongly suppressed evaporation in tropical South America, Africa, and Southeast Asia (Fig. 3). This robust spatial pattern suggests that the relative humidity reductions are driven by the $CO_2$ physiological effect, a mechanism included in the land models of 11 GeoMIP simulations, all but EC-Earth (Table 1). In response to elevated ambient $CO_2$ concentrations, plants constrict their stomata, which reduces evapotranspiration in the high $CO_2$ simulations, including the G1 simulations (Kravitz et al., 2013b; Cao et al., 2010). In the global warming (abrupt4xCO2) CMIP5 simulations, this effect

is partially offset by the increased net primary productivity in a warmer world. However, in G1, this net primary productivity effect is muted by the reduction in insolation. Tilmes et al. (2013) found that the plant physiological response in G1 is qualitatively the same as for abrupt4xCO2. In another study, biogeochemical cycling was found to influence global precipitation as much as the radiative reduction itself (Fyfe et al., 2013).

  In the National Center for Atmospheric Research (NCAR) Community Land and Community Atmosphere Model, Cao et al.

(2010) isolated the $CO_2$ physiological effect from a doubling of atmospheric $CO_2$. They reported patterns of reduced latent heat flux and relative humidity from this vegetative forcing that closely resemble those we observe in the GeoMIP suite, in Fig. 3 and Fig. 5. This is also consistent with the reasoning of Bala et al. (2008) in that the surface energy budget constrains the response to the shorwave forcing of the G1 experiment. When the downward shortwave flux decreases, the surface fluxes must respond, and in this case the latent heat flux dominates the response. Evaporation decreases, and precipitation follows (Fig. 3).

In the present study, since strong and significant reductions in relative humidity over land are largely constrained to regions

with extensive vegetation in the form of boreal, temperate or tropical forests, we consider the biogeochemical effect of $CO_2$ to be the dominant cause of the relative humidity change. The role of these local $H_s$ changes is minimal in zonal mean climate (Fig. 4C), but the gradient of the changes in $H_s$ could be responsible for some of the simulated $P - E$ changes, particularly at smaller spatial scales, such as over sub-Saharan Africa and the tropical rainforests of South America. We leave investigation of this effect to future research.

## 2.3 Dynamically Driven Precipitation

The third factor we consider in decomposing the $P - E$ response to geoengineering is the atmospheric circulation. Large-scale meridional circulations are driven by energy gradients imposed by the uneven distribution of sunlight on Earth. The Hadley circulation cells are responsible for most of the poleward heat transport in the tropics, where the annual solar input is highest (Hill et al., 2015). The net energy flux of the Hadley circulation is in the flow direction of its upper branch (Held, 2001). The ascending motion of the Hadley cell drives the seasonally-migrating tropical rainfall known as the ITCZ, and there is evidence that its position is determined by meridional gradients in the vertically-integrated atmospheric energy budget (Shekar and Boos, 2016). The Hadley circulation is crucial for balancing global energy, so high-latitude temperature anomalies can drive shifts of the ITCZ (Yoshimori and Broccoli, 2008). The ITCZ is sensitive to interhemispheric energy contrasts set up by aerosols, clouds, or antisymmetric heating (Seo et al., 2014). A thorough analysis of Hadley circulation changes is a crucial outstanding task for understanding the hydrological response to solar geoengineering (Kravitz et al., 2013b). We will quantify changes to the Hadley circulation with the meridional streamfunction. The meridional streamfunction is derived from the continuity equation, and either $\bar{v}$, the meridional wind vector, or $\bar{w}$, the vertical wind vector, can be used to fully define the two-dimensional, overturning flow (Eq. (4)):

$$\Psi(\phi, p) = 2\pi a \cos\phi \int\limits_0^p \bar{v}\, \mathrm{d}p/g. \tag{4}$$

where $\phi$ is the latitude, $p$ is pressure, $a$ is the Earth's radius, $\bar{v}$ is the meridional velocity, and $g$ is gravity.

Changes in top of atmosphere (TOA) energy fluxes influence the direction and strength of ITCZ shifts (Kang et al., 2008). Numerous studies have noted the strong relationship between ITCZ position and the hemispheric temperature contrast as well. The correlation between interhemispheric temperature contrasts and annual mean ITCZ position is a robust result and is related to extratropical energy transport (e.g., Broccoli et al., 2006; Toggweiler and Lea, 2010). Schneider et al. (2014) explain how this is consistent with an energetic framework: the hemisphere with the higher average temperature typically has a smaller meridional temperature gradient due to the near symmetry of tropical temperatures about the equator. This corresponds to reduced poleward extratropical eddy transport in that hemisphere, and increased energy flux by the atmosphere across the equator and out of the hemisphere by the upper branch of the Hadley cell. The ITCZ is drawn towards the warmed hemisphere because moisture is transported in the opposite direction as energy by the Hadley cell. Therefore, we investigate the possibility

that differing dynamical responses to solar dimming among the models are due to differences in the temperature restoration of the Northern and Southern Hemispheres.

To discern the component of the precipitation change caused by changes in large scale atmospheric dynamics, we calculated the change in the Hadley circulation between the G1 (years 11-50) and piControl (final 40 years) simulations. For each model, we computed the meridional streamfunction over this 40 year averaging period based on the modeled meridional wind vector, as in Eq. (4). Data were unavailable for the CESM and HadC models. We examined annual and seasonal mean dynamical changes to understand the response of the zonal mean hydrological cycle, including the periods July-August-September (JAS) and January-February-March (JFM). (We chose these averaging periods because the multi-model mean ITCZ position extremes occur in August and February.) To better interpret the dynamical changes, we assessed the annual mean and seasonal changes in the interhemispheric temperature contrast between G1 and piControl for each model by calculating area-weighted hemispheric averages of the surface temperature, averaged over a 40 year period (years 11-50 of G1 and 1-40 of piControl). The ITCZ shift between G1 and piControl is defined as the shift of the precipitation centroid. This is the latitude between 15°N and 15°S at which half the precipitation is to the north and half is to the south.

The annual mean Hadley circulation changes vary in magnitude and direction amongst the GeoMIP ensemble members and contribute to dynamic moistening and drying. The meridional streamfunction plots suggest that the northward (HadGEM2-ES) and southward (GISS-E2-R, MIROC-ESM) ITCZ shifts, characterized by counterclockwise or clockwise tropical anomalies respectively, are dynamically driven (Fig. 6). The anomalous ascent at the equator in CCSM4 and NorESM1-M accounts for the narrowing of the ITCZ noted in the zonal mean $P - E$ figure. The mean circulation does not seem to provide a dynamical basis for the annual mean constriction of the ITCZ in the MPI-ESM-LR and IPSL-CM5A-LR models, in which anomalies are less than $10^{10}$ kg s$^{-1}$. Small changes in the latitudinal range and strength of the Hadley circulation and associated precipitation have large local implications, especially on subannual scales (Kang et al., 2009). Boreal summer (JAS) and winter (JFM) meridional streamfunction anomalies are in every model stronger than the annual mean (Figs. 7, 8). In HadGEM2-ES, for example, the JAS meridional mass flux anomaly exceeds $4 \times 10^{10}$ kg s$^{-1}$. On the opposite extreme, the IPSL-CM5A-LR model JAS and JFM mass flux anomalies are below $1.5 \times 10^{10}$ kg s$^{-1}$. In general, the JAS streamfunction changes rather than the JFM anomalies set the pattern for the annual mean circulation change (Figs. 6-8). In the JAS average, there is anomalous energy transport toward the summer hemisphere (NH) in eight of nine models (all but HadGEM) (Fig. 7). In the JFM average, there is again anomalous energy transport toward the summer hemisphere (SH), though the result is less consistent across the suite (seven of nine models) (Fig. 8). These changes in the Hadley cell mass flux are consistent with the relative cooling of the summer hemisphere throughout the year (Fig. 9b,c).

We find that the shifts of annual mean tropical rainfall in the models are correlated with the interhemispheric surface temperature contrasts (Correlation coefficient $(r) = 0.71$, Fig. 9a). Models with higher annual mean surface temperatures in the Northern Hemisphere under geoengineering tend to display northward shifts of the ITCZ. This is consistent with previous research that shows a strong relationship between the ITCZ position and the hemispheric temperature contrast (e.g., Kang et al., 2008; Frierson and Hwang, 2012). Modeling studies by Haywood et al. (2013, 2016) have shown that increasing the albedo by injecting stratospheric aerosols into only one hemisphere could cause substantial shifts in the ITCZ toward the other

hemisphere. Our analysis of the G1 experiment suggests that similar effects could occur, albeit on a smaller scale, even with a hemispherically symmetric injection strategy, which is approximated by reducing the solar constant. Despite the hemispherically symmetric forcing induced by solar dimming, the ensemble mean residual high-latitude warming is larger in the Arctic than in the Antarctic (Fig. 1), and in 9 out of 11 models the Northern Hemisphere is warmed relative to the Southern Hemisphere after geoengineering (Fig. 7a). This suggests that there could be an intriguingly close relationship between the degree of Arctic warming amplification and the tropical hydrological response to geoengineering in models. The relationship between ITCZ shifts and energy transport in G1 will be further explored in a future study.

One response of the ITCZ to the G1 experiment that is consistent across all 11 models is that the seasonal migration of the ITCZ is dampened. Figure 10 shows the annual, boreal winter (JFM), and boreal summer (JAS) mean position of the ITCZ in each model in piControl (years 1-40) and G1 (years 11-50). In each model, the distance between the seasonal mean positions of the ITCZ is reduced. In some models there is a poleward shift in the ITCZ in one of the seasons, but in each of these cases there is a greater equatorward shift in the opposite season, with an annual mean ITCZ shift and a reduction in the seasonal migration occurring simultaneously.

The reduction in the seasonal ITCZ migration is consistent with the physical mechanism relating sulfate aerosols and ITCZ shifts during 1971-1990 described by Hwang et al. (2013; see their Figure 4). There is more available sunlight in the summer hemisphere, which results in a greater cooling there when the solar constant is reduced. To compensate for the loss of energy in the summer hemisphere, the climatological energy flux out of the summer hemisphere and towards the winter hemisphere is reduced. Indeed, in G1, most models show an anomalous Hadley circulation in which winds aloft, and therefore energy, move towards the summer hemisphere (Figs. 7, 8). This is accompanied by anomalous flow towards the winter hemisphere in the lower branch of the Hadley cell, which weakens moisture transport towards the summer hemisphere and moves the summer ITCZ position away from the summer pole. The warming of the winter relative to the summer hemisphere and the ITCZ shift toward the winter hemisphere are correlated between the different models (Fig. 9b,c), and are consistent with the proposed physical mechanism.

Damped seasonal ITCZ migration provides a possible physical mechanism for the narrowing of the annual mean ITCZ in the various models. Other processes that could affect the width of the ITCZ include changes in gross moist stability, the net energy input to the atmosphere, or the advection of moist static energy by the Hadley cell mean flow or transient eddies, as analyzed by Byrne and Schneider (2016a; 2016b) for global warming simulations. A more comprehensive analysis of the processes responsible for the contraction of tropical precipitation in solar geoengineering experiments would be a useful avenue for future study. Reduced seasonal ITCZ migration due to summer hemisphere cooling is also one possible physical explanation for the reduction in summer monsoon precipitation in the G1 experiment found by Tilmes et al. (2013).

## 3   Conclusions

Hadley circulation changes play a significant role in driving the $P - E$ changes in climate model simulations of uniform solar dimming. While thermodynamic scaling captures the general spatial structure of $P - E$ changes under global warming, it does

not do so for idealized simulations of the response to increased $CO_2$ combined with solar geoengineering. Hadley circulation changes are in qualitative agreement with zonal mean features of the hydrological response to G1, so we conclude that they play a primary role in the response. Thermodynamic scaling and relative humidity changes may be important to explain $P - E$ anomalies over rainforests or at high latitudes where the $CO_2$ physiological response and residual temperature anomalies are more important, respectively.

The models can be divided into three groups characterized by different tropical $P - E$ responses to geoengineering: either a southward shift, northward shift, or narrowing of the ITCZ. Our results support that changes in tropical dynamics, namely shifts of the Hadley circulation, are largely responsible for these alterations to the $P - E$ distribution. In a previous study, convection scheme parameters were determinative of the tropical precipitation response to extratropical forcings (Kang et al., 2009), and other studies (Song and Zhang, 2009; Liu et al., 2010) have also found tropical precipitation to be sensitive to the convection scheme. The partitioning of cross-equatorial fluxes between atmospheric and oceanic components is also important for the resulting ITCZ shift, so differences in the oceanic component of the models could emerge as significant (Kang et al., 2008).

We also present evidence that land-sea contrasts in evaporation rates, resulting in land-sea contrasts in relative humidity anomalies, may contribute to small changes in $P - E$ with solar dimming. We do so by examining the spatial distribution of relative humidity and evaporation anomalies, and by calculating an extended thermodynamic scaling that accounts for the response of $P - E$ to the local relative humidity change. We reason that these relative humidity changes are related to the effect of $CO_2$ on the stomatal conductance in plants, a phenomenon noted in previous studies of geoengineering. It would be interesting to examine the relative humidity anomalies in G1 on shorter temporal scales given the important role of vegetation, a seasonally varying feature of the climate, in its modulation. It would also be valuable to analyze the influence of gradients of changes in relative humidity on $P - E$, as this was shown to be an important influence over land in previous work by Byrne and O'Gorman (2015).

Tropical precipitation is sensitive to solar perturbations and would be altered by an implementation of globally-uniform solar geoengineering. Based on our inter-model comparison, there is substantial uncertainty regarding the nature of the tropical precipitation response, in terms of the direction and strength of the ITCZ shift, as well as its variation on seasonal time scales. We present evidence that residual warming of one hemisphere relative to the other under geoengineering draws annual mean tropical rainfall into that hemisphere. On seasonal timescales, preferential cooling of the summer hemisphere results in a damping of the seasonal migration of the ITCZ, which may help explain the apparent narrowing of the tropical peak in annual mean precipitation. A reduced seasonal migration of the ITCZ could have catastrophic consequences for semi-arid areas like the Sahel region, which lies at the northern margin of the current seasonal ITCZ excursion. The potential for such regional impacts under geoengineering warrants further study. Our results reinforce the finding that uniform solar dimming cannot restore preindustrial conditions in terms of $P - E$ patterns, a fundamental aspect of climate. An investigation of the ability of spatially targeted solar geoengineering to offset these $P - E$ changes would be a valuable future direction. In light of the considerable inter-model differences, improvements in model representation of processes including clouds and tropical convection will also

help improve our understanding of hydrological cycle responses to solar geoengineering.

*Author contributions.* T. Storelvmo designed research and J.E. Smyth performed the analysis. J.E. Smyth and T. Storelvmo interpreted results, and J.E. Smyth wrote the manuscript with input from the coauthors. R.D. Russotto contributed Figures 9 and 10 and wrote several paragraphs discussing them.

*Acknowledgements.* Four anonymous reviewers provided comments which helped to improve the manuscript. We thank the climate modeling groups for participating in the Geoengineering Model Intercomparison Project and for making their data available. In particular we thank Dr. Ben Kravitz and the scientists managing the Earth System Grid Federation for facilitating data access. T. Storelvmo was supported by NSF under grant 1352417. J.E. Smyth was supported by the Karen Von Damm '77 Undergraduate Research Fellowship from the Yale University Department of Geology & Geophysics. R.D. Russotto was supported in part by the U.S. Department of Defense (DoD) through the National Defense Science and Engineering Graduate Fellowship (NDSEG) Program.

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

FIGURES

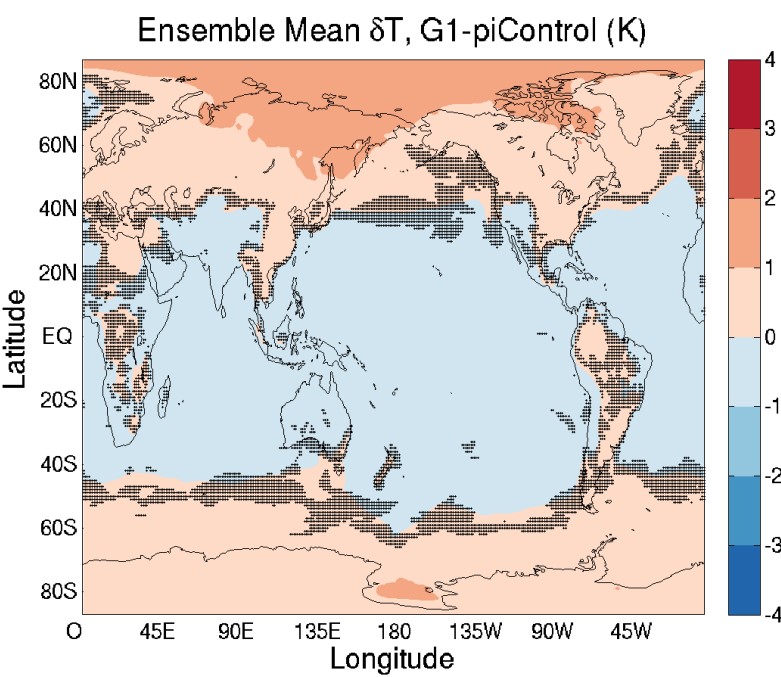

**Figure 1.** The annual mean distribution of near-surface atmospheric temperature anomalies (K) between G1 (years 11-50) and piControl (all years). Stippling denotes regions where fewer than 66% of the 12 ensemble members agree on the sign of the change. These results appear in Kravitz et al. 2013a.

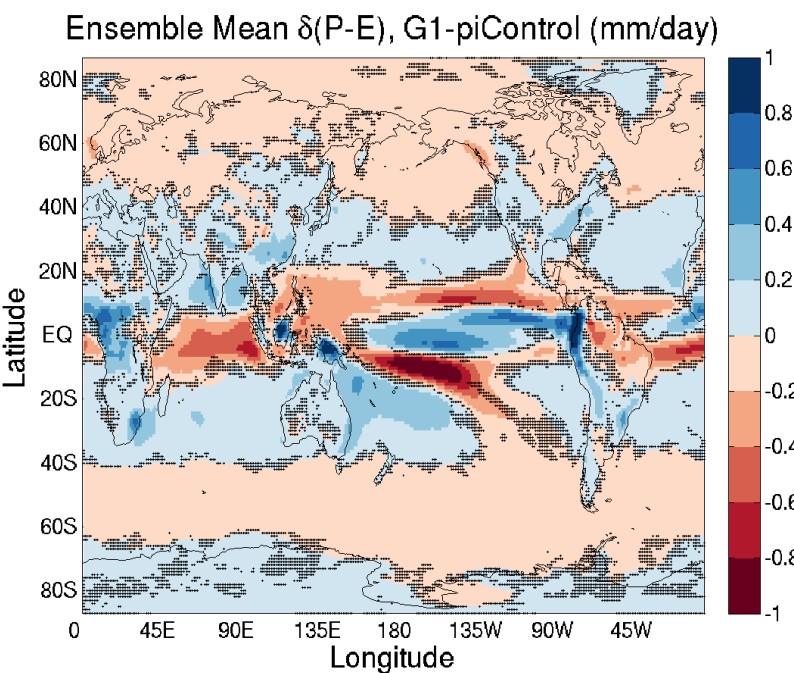

**Figure 2.** The annual mean distribution of precipitation minus evaporation rate anomalies (mm/year) between G1 (years 11-50) and piControl (all years), averaged among 11 models (EC-Earth excluded due to unphysical result). Stippling indicates where fewer than 64% of models agree on the sign of the change. These results appear in Kravitz et al. 2013a.

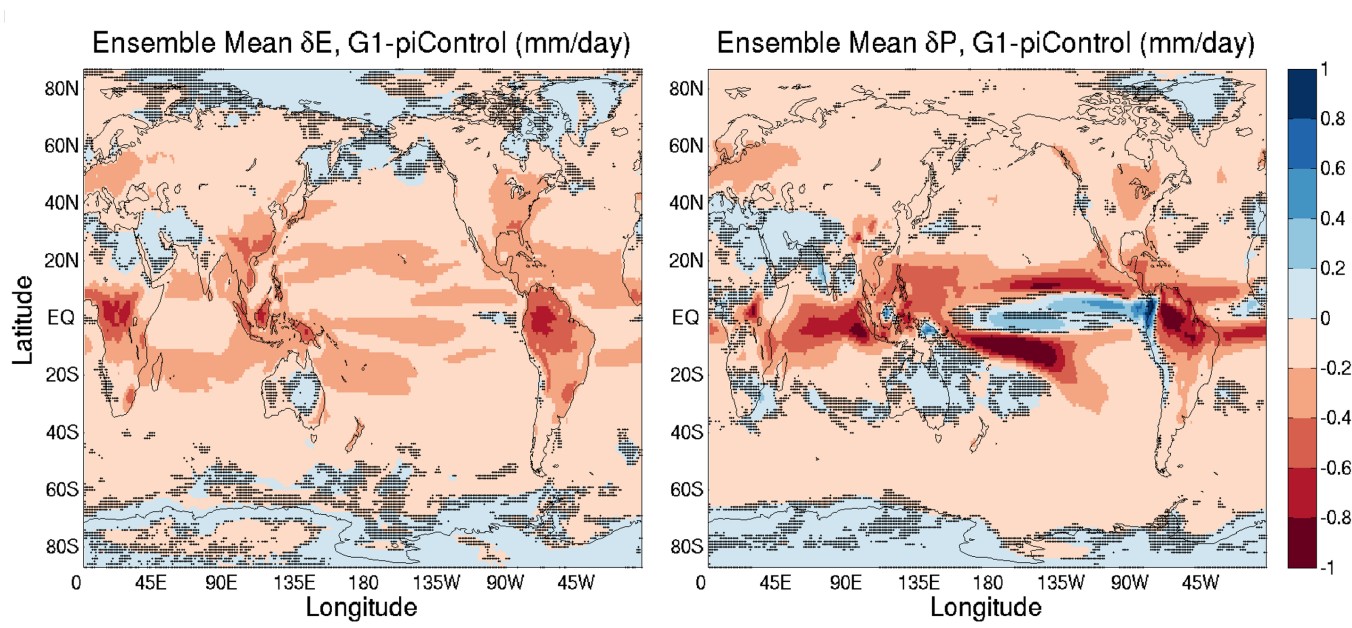

**Figure 3.** The annual mean distribution of evaporation (left panel) and precipitation (right panel) rate anomalies (mm/year) between G1 (years 11-50) and piControl (all years), averaged among 11 models (EC-Earth excluded due to unphysical result). Stippling indicates where fewer than 64% of models agree on the sign of the change. These results appear in Kravitz et al. 2013a.

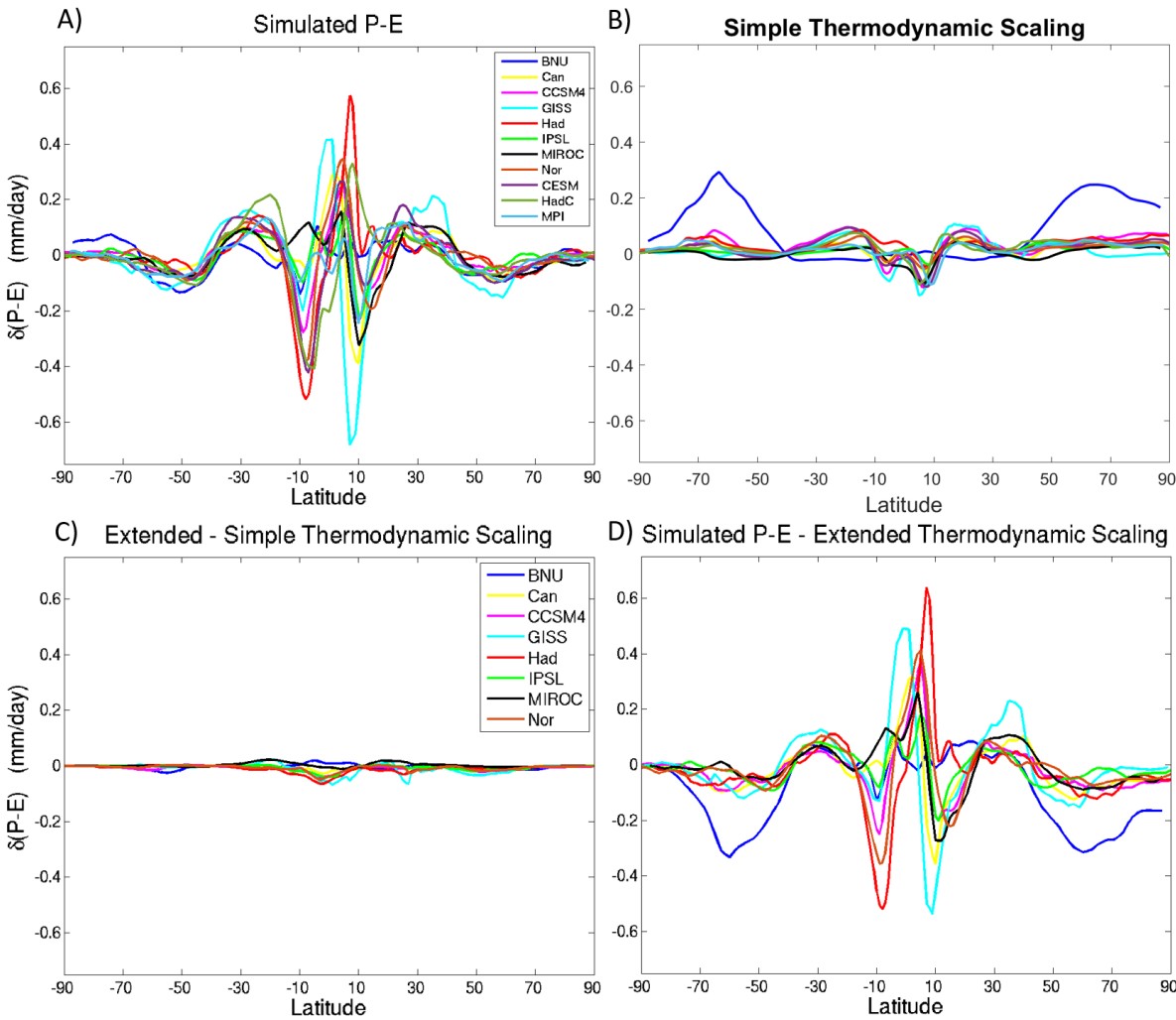

**Figure 4.** A) shows the zonal mean $\delta(P-E)$ for G1-piControl simulated in 11 climate models, and B) is the $P-E$ anomaly predicted by the simple thermodynamic scaling in Eq. (2). C) shows the difference of the G1-piControl $\delta(P-E)$ predicted by the extended (Eq. (3)) and simple (Eq. (2)) scalings. This isolates the contribution of local relative humidity changes to the $P-E$ anomalies. D) is the difference between the simulated G1-piControl $\delta(P-E)$ and the $P-E$ anomaly predicted by the extended scaling, and represents the changes in dynamically driven rainfall. (EC-Earth excluded due to unphysical result).

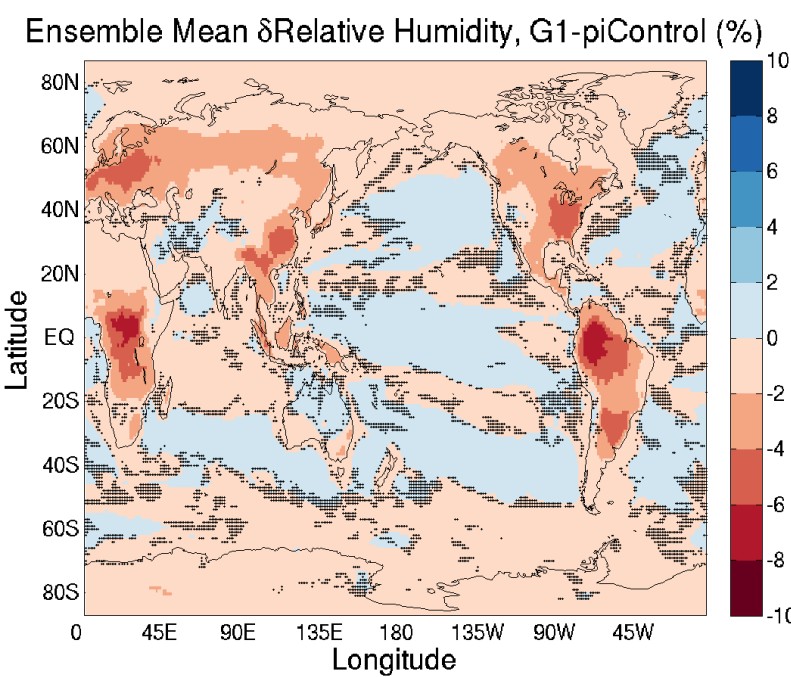

**Figure 5.** The annual mean near-surface relative humidity anomaly between G1 (years 11-50) and piControl (all years) in eight GCMs. Stippling indicates that fewer than 62.5% of the models agree on the sign of the change. (Data unavailable for HadC, CESM, and MPI models; EC-Earth excluded).

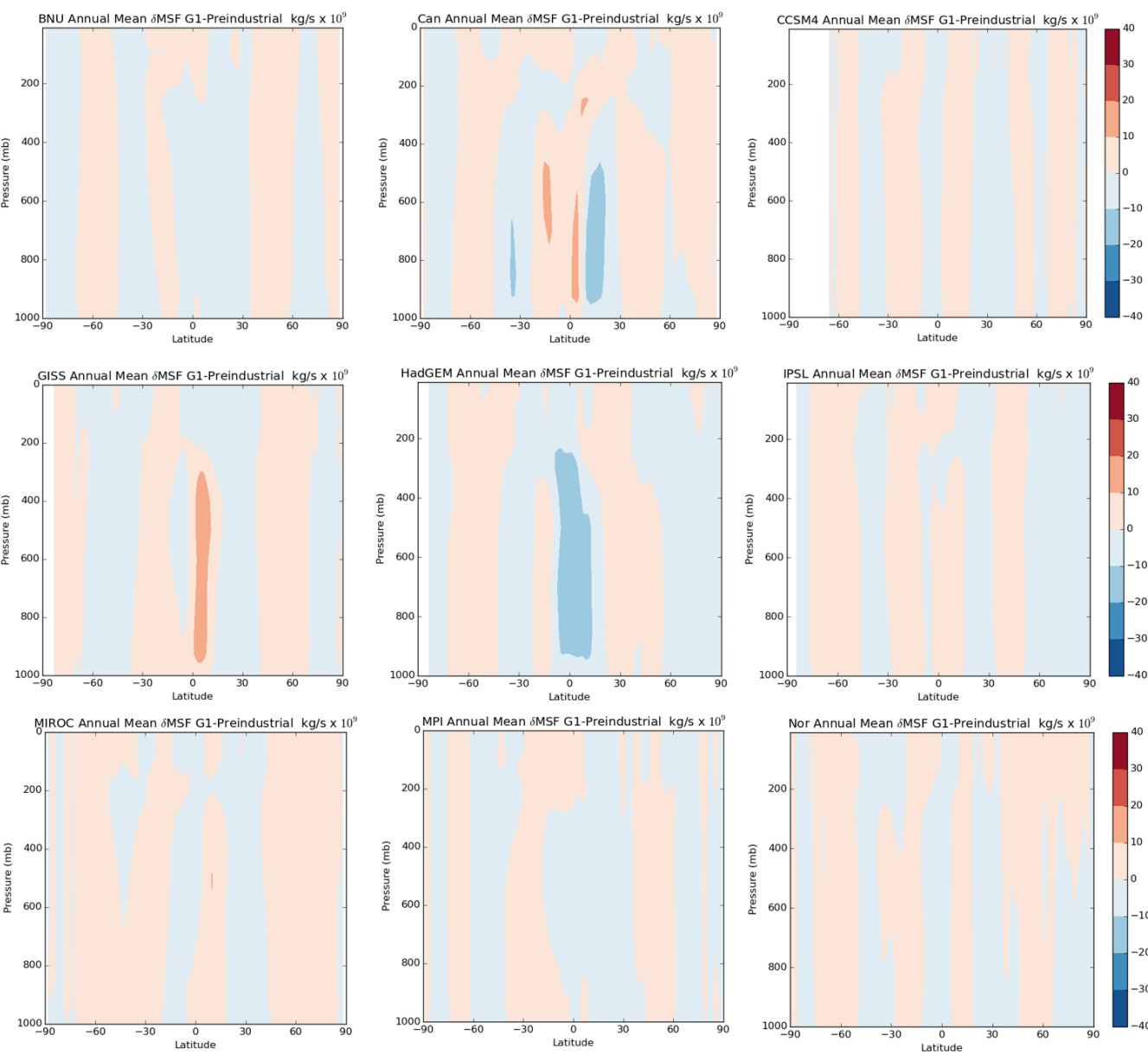

**Figure 6.** The annual mean meridional streamfunction anomaly between G1 (years 11-50) and piControl (last 40 years) in each model, as calculated in Eq. (4). Blue colors indicate counterclockwise motion. (Data unavailable for HadC and CESM models).

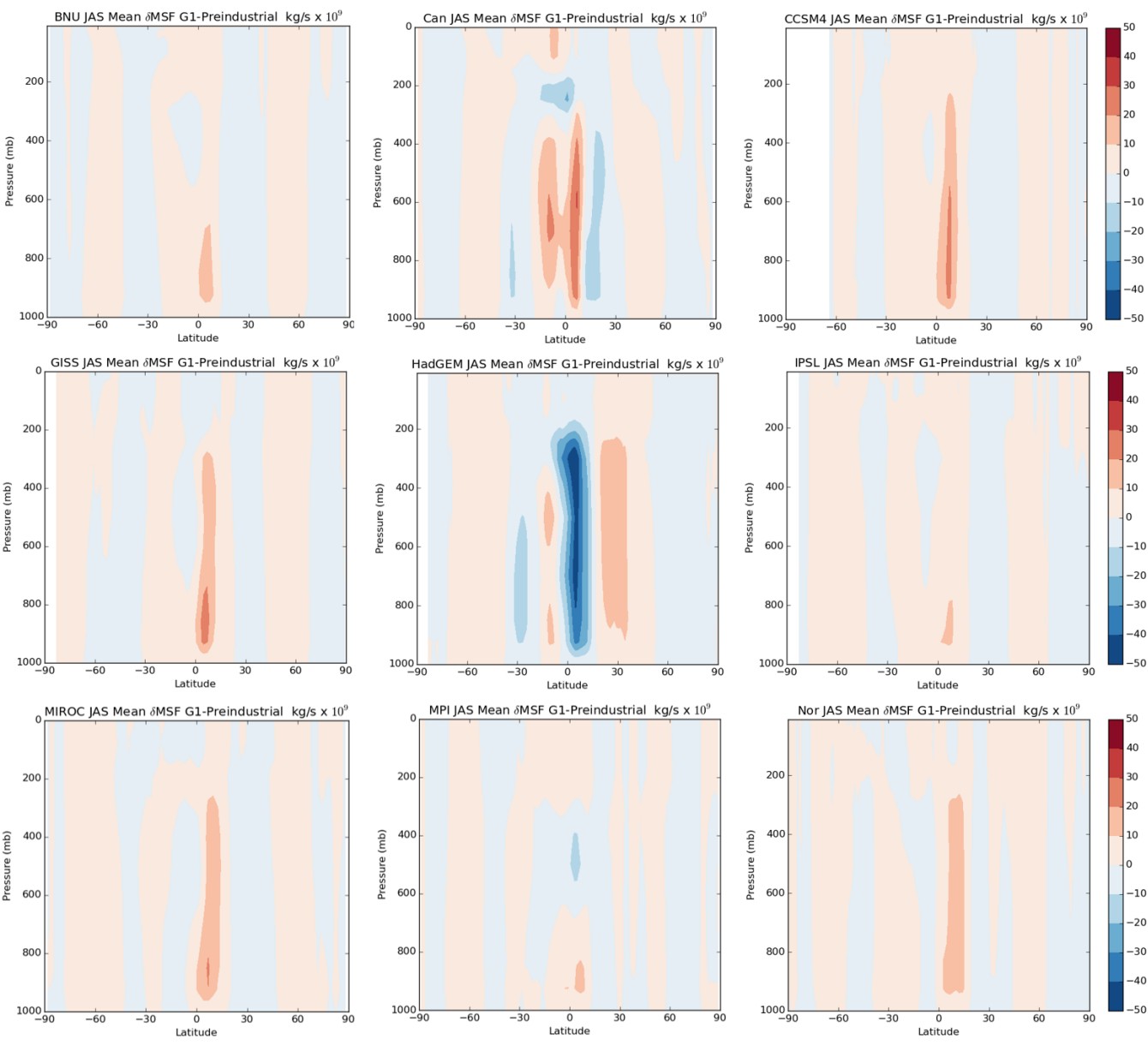

**Figure 7.** The JAS mean meridional streamfunction anomaly between G1 (years 11-50) and piControl (last 40 years) in each model, as calculated in Eq. (4). Blue colors indicate counterclockwise motion. (Data unavailable for HadC and CESM models).

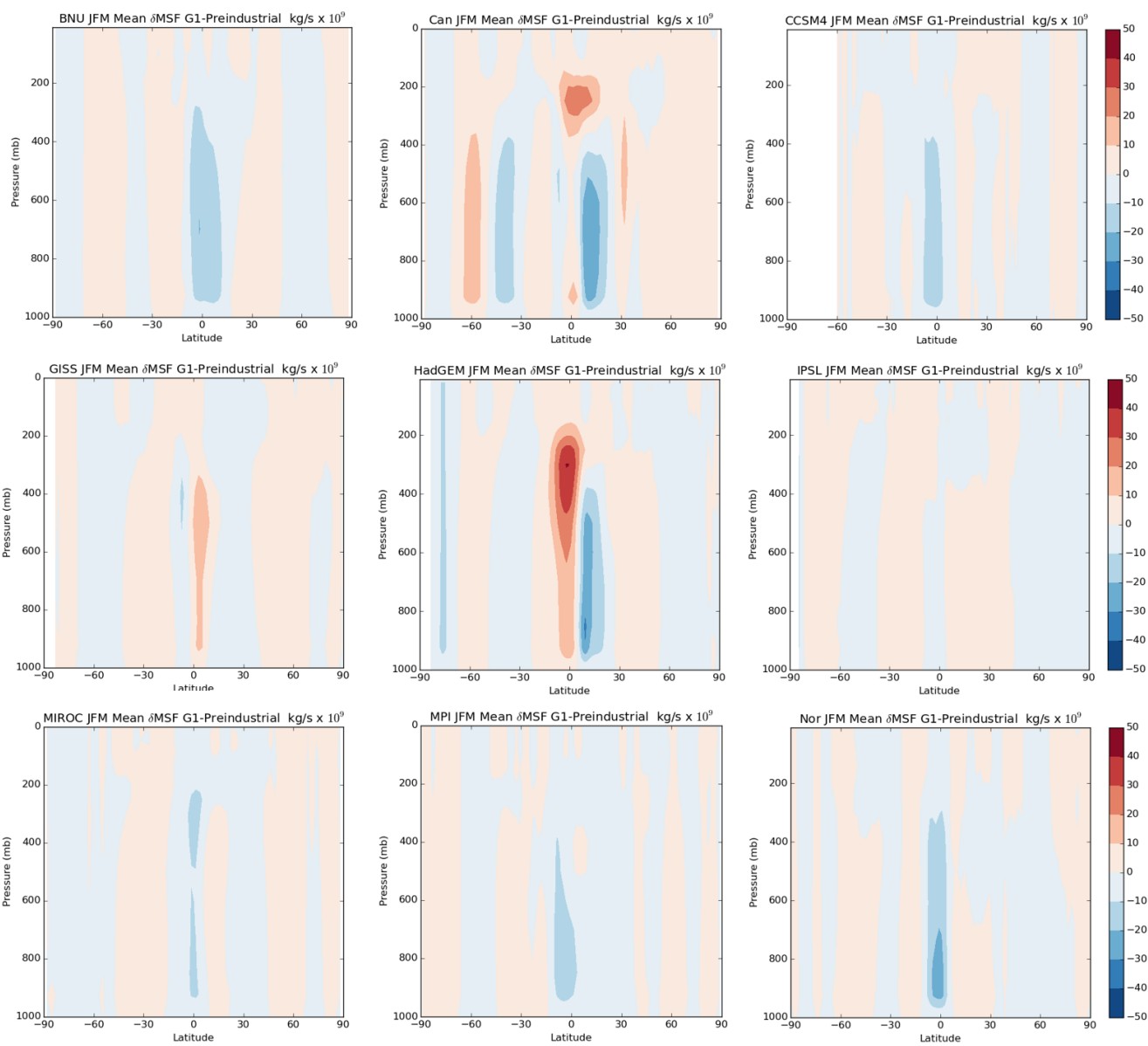

**Figure 8.** The JFM mean meridional streamfunction anomaly between G1 (years 11-50) and piControl (last 40 years) in each model, as calculated in Eq. (4). Blue colors indicate counterclockwise motion. (Data unavailable for HadC and CESM models).

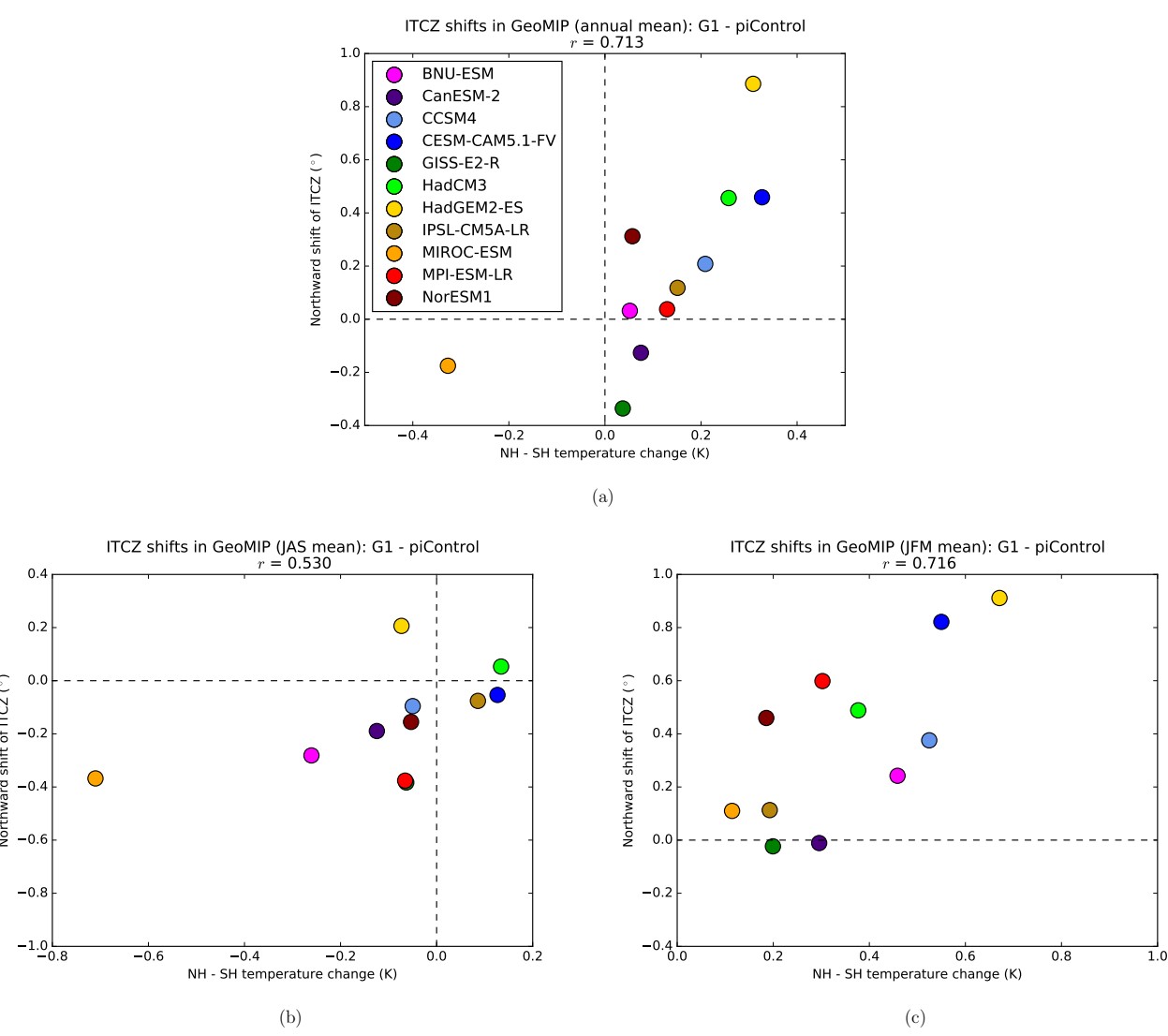

**Figure 9.** The ITCZ shift vs. the anomaly of the interhemispheric temperature contrast between G1 (years 11-50) and piControl (years 1-40), where $r$ is the correlation coefficient. Panel a) shows the annual mean, b) is the JAS mean, and c) is the JFM mean.

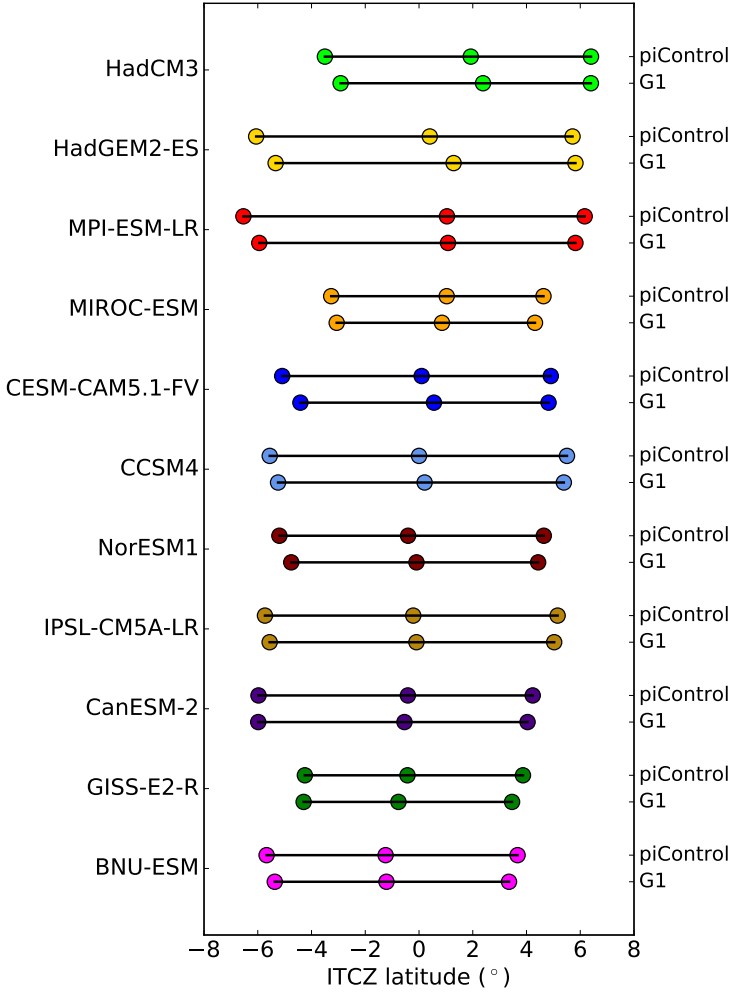

**Figure 10.** Annual and seasonal mean positions of the ITCZ in piControl (years 1-40) and G1 (years 11-50). For each model, the top row of dots shows piControl positions, and the bottom row of dots shows G1 positions. In each row of dots, the left dot shows the JFM mean position, the middle dot shows the annual mean position, and the right dot shows the JAS mean position. Models are ordered by the annual mean ITCZ position in piControl.

**Table 1.** GeoMIP Model Specifications. In certain figures models are labeled with the shortened name in parenthesis. Column 3 refers to the $CO_2$ physiological effect in plants. The solar constant ($S_0$) reduction is a percentage. Information courtesy of Kravitz et al. (2013a)

| Model[1] | Dynamic Vegetation | Phys. Effect | $S_0$ Reduction | References |
|---|---|---|---|---|
| BNU-ESM (BNU) | no | yes | 3.8 | Ji et al. (2014) |
| Can-ESM2 (Can) | yes | yes | 4.0 | Arora et al. (2011) |
| CCSM4 (CCSM4) | no | yes | 4.1 | Gent et al. (2011) |
| CESM-CAM5.1-FV (CESM) | no | yes | 4.7 | Hurrell et al. (2013) |
| EC-Earth | no | no | 4.3 | Hazeleger et al. (2012) |
| GISS-E2-R (GISS) | no | yes | 4.5 | Schmidt et al. (2014) |
| HadCM3 (HadC) | no | yes | 4.1 | Gordon et al. (2000) |
| HadGEM2-ES (Had) | yes | yes | 3.9 | Collins et al. (2011) |
| IPSL-CM5A-LR (IPSL) | yes | yes | 3.5 | Dufresne et al. (2013) |
| MIROC-ESM (MIROC) | yes | yes | 5.0 | Watanabe et al. (2011) |
| MPI-ESM-LR (MPI) | no | yes | 4.7 | Giorgetta et al. (2013) |
| NorESM1-M (Nor) | no | yes | 4.0 | Bentsen et al. (2013) |

**1. Full Names:** BNU-ESM, Beijing Normal University-Earth System Model; CanESM2, The Second Generation Canadian Earth System Model; CESM-CAM5.1, The Community Climate System Model Version 5.1; CCSM4, The Community Climate System Model Version 4; EC-EARTH DMI, European Earth System Model based on ECMWF Models (Seasonal Forecast System), Danish Meteorological Institute; GISS-E2-R, Goddard Institute for Space Studies ModelE version 2; HadCM3, Hadley Centre coupled model 3; IPSL-CM5A-LR, Institut Pierre Simon Laplace ESM; MIROC-ESM, Model for Interdisciplinary Research on Climate-Earth System Model; MPI-ESM-LR, Max Planck Institute ESM; NorESM1-M, Norwegian ESM.