# Peer review of "Thermodynamic and dynamic responses of the hydrological cycle to solar dimming"

_Atmospheric Chemistry and Physics, 2016_

## Referee Comment (RC1) · Anonymous Referee #1 · 19 Oct 2016

Review of Smyth et al. acp-2016-886

In general, I think this is a nice paper. The study is clean and straightforward, and I think we've learned something new. I do have a few comments:

General comments:

1. I would periodically get lost in the paper and had to re-read quite a number of lines. It would help if there were an outline at the beginning (maybe even a numbered list) of the factors that affect hydrological cycle changes. Then you can go through them one by one.

2. The abstract doesn't really say much, and the conclusions don't appear to agree with the results presented. Maybe I'm confused somewhere, in which case I think the

description needs to be clearer.

Specific comments:

Page 1, line 24: Not all of the models were run for 500 years after spin-up. This really varies among the different modeling groups.

Page 2, line 8: Why only 12? You should say more about this here.

Page 2, lines 24ff: This isn't a sentence.

Page 3, lines 1ff: Somewhere in here, you should discuss how well these assumptions hold, and if they don't, what you can still learn.

Page 3, line 6: Instead of "project", use "study" or something like that.

Page 4, lines 1-2: I don't think this is quite fair. They had a lot of warming at high latitudes in particular, so it makes sense there would be a P-E responds, regardless of whether the temperature response to 4xCO2 is compensated.

Page 5, line 22: Typo - two periods at the end of the sentence.

Page 6, lines 14 and 16: Citation is coming out weirdly. Put 2014 on line 14.

Page 7, line 21: This doesn't strike me as consistent with what you discussed earlier, nor is it what figure 4 shows.

Page 8, lines 6-7: This is not correct as written. Turning down the sun by a uniform fraction cannot restore preindustrial P–E patterns. That doesn't say anything about geoengineering as a whole.

Figure 7: The paper by Haywood et al., 2015, GRL might be relevant here. (You don't need to do anything about this unless you want to – I just thought you'd find it interesting.)

---

## Referee Comment (RC2) · Anonymous Referee #2 · 30 Oct 2016

I recommend this paper be rejected. It presents little new information. It repeats results from previous papers. And it ignores the seasonal cycle in precipitation and evaporation which includes a lot of physics and monsoon responses, as analyzed by Tilmes et al. (2013). The conclusions are either obvious or not sufficiently diagnosed to add to understanding.

There are many comments in the attached annotated manuscript that need to be addressed. In addition:

I am confused. The text says there were 13 models and you excluded one, but do not say which model and why. Then Figs. 2 and 3 used 11 models, but excluded one. Again, what was the criterion for excluding the model? Table 1 only lists 12 models.

Fig. 1 is not a new result. It is the same as Fig. 2 (top right) of Kravitz et al. (2013a),

and this needs to be acknowledged.

Figs. 2 and 3 are not new. They are the same as Fig. 5 of Kravitz et al. (2013a), and this needs to be acknowledged.

I don't understand which results are plotted in Fig. 4A. Is it G1? piControl? The difference? The caption just says "as simulated in the models." If it is the difference, why does it differ from the results shown in Fig. 1 of Kravitz et al. (2103a), and again not acknowledged?

Graphics are poor quality. For Figs. 1-3, 5: - The color shading has way too many shades, so it is impossible to determine the value by looking at a color on the map. Use fewer values and include labeled contour lines. - The stippling is much too dense. It is impossible to see the shading underneath it. - The x-axis label is wrong. The scaling is wrong and the longitude labels are in the wrong place. The right end should be $0°$. - The entire figure is blurry and too low resolution. - The criterion for shading varies from 62.5% to 64% to 66% agreement. Why? Why not use the 75% criterion of Kravitz et al. (2013a), which covers less of the data? - Try using GrADS. It makes beautiful maps, and automatically gives you labeling, contours, and shading.

For Fig. 6, the color shading has way too many shades, so it is impossible to determine the value by looking at a color on the map. Use fewer values and include labeled contour lines.

Use "piControl" rather than "Preindustrial," as it is the standard CMIP5 terminology.

Please also note the supplement to this comment:
http://www.atmos-chem-phys-discuss.net/acp-2016-886/acp-2016-886-RC2-supplement.pdf
* * *
[Figure]

**Supplement:**

[revised manuscript text omitted]

---

## Author Comment (AC1) · 13 Jan 2017

**acp-2016-886: Thermodynamic and dynamic responses of the hydrological cycle to solar dimming**

Reviewer Comments are written in black font, and the Author Responses are in blue.

**Response to RC1**

*General comments*:

1. I would periodically get lost in the paper and had to re-read quite a number of lines. It would help if there were an outline at the beginning (maybe even a numbered list) of the factors that affect hydrological cycle changes. Then you can go through them one by one.

Thank you for the suggestion. At the end of the Introduction (Section 1), we have added a bulleted list of the factors we investigate in subsequent sections. We also added opening sentences to Sections 2.1, 2.2., and 2.3 to reorient the reader.

2. The abstract doesn't really say much, and the conclusions don't appear to agree with the results presented. Maybe I'm confused somewhere, in which case I think the description needs to be clearer.

We have revised the abstract to better summarize the results. Thank you for the advice.

Specific comments:

Page 1, line 24: Not all of the models were run for 500 years after spin-up. This really varies among the different modeling groups.

We have deleted this sentence, as the protocol for the GeoMIP experiments is addressed in Kravitz et al. 2010, which we cite.

Page 2, line 8: Why only 12? You should say more about this here.

We now explain this in the text.

Page 2, lines 24ff: This isn't a sentence.

The sentence is no longer interrupted by Equation (1) and reads more clearly now.

Page 3, lines 1ff: Somewhere in here, you should discuss how well these assumptions hold, and if they don't, what you can still learn.

We have added discussion of these assumptions, which better frames the subsequent adaptation of Byrne and O'Gorman's "extended scaling," which accounts for the fact that an assumption of constant relative humidity breaks down over land.

Page 3, line 6: Instead of "project", use "study" or something like that.

This has been changed.

Page 4, lines 1-2: I don't think this is quite fair. They had a lot of warming at high latitudes in particular, so it makes sense there would be a P-E responds, regardless of whether the temperature response to 4xCO2 is compensated.

We agree that it is unsurprising that the thermodynamic scaling calculation shows a P-E increase at high latitudes in BNU-ESM, due to the particularly large positive G1-Preindustrial temperature anomalies in this model (Fig. 4B). We have clarified these sentences.

Page 5, line 22: Typo - two periods at the end of the sentence.

This is now corrected.

Page 6, lines 14 and 16: Citation is coming out weirdly. Put 2014 on line 14.

Thank you. We have corrected this.

Page 7, line 21: This doesn't strike me as consistent with what you discussed earlier, nor is it what figure 4 shows.

Thank you for drawing attention to this. We have changed the paragraph to convey the relative importance of the three mechanisms we assessed in the study. The main driver of P-E changes under solar dimming is the Hadley circulation.

Page 8, lines 6-7: This is not correct as written. Turning down the sun by a uniform fraction cannot restore preindustrial P–E patterns. That doesn't say anything about geoengineering as a whole.

We have revised the end of this paragraph to reflect this distinction.

Figure 7: The paper by Haywood et al., 2015, GRL might be relevant here. (You don't need to do anything about this unless you want to – I just thought you'd find it interesting.)

Thank you for the suggestion, as the work of Haywood et al. is indeed relevant. We have now cited 2 papers by Haywood et al. (2013, Nature Climate Change; and 2016, GRL) in the fourth to last paragraph of Section 2.3.

**Response to RC2**

I recommend this paper be rejected. It presents little new information. It repeats results from previous papers. And it ignores the seasonal cycle in precipitation and evaporation which includes a lot of physics and monsoon responses, as analyzed by Tilmes et al. (2013). The conclusions are either obvious or not sufficiently diagnosed to add to understanding.

We disagree with your assessment of the paper. No previous study has decomposed the hydrological response to geoengineering into thermodynamic, relative humidity-driven, and dynamical components. In fact, no previous study has plotted the Hadley streamfunction

changes in the G1 experiment, and this was identified as a valuable future research direction by Kravitz et al. 2013c, as noted in our Introduction. Thermodynamic scaling captures the general spatial structure of P-E changes under global warming, as discussed in Held and Soden (2006), but we find that it is less explanatory of the hydrological response to geoengineering. Relative humidity changes over land are substantial in GeoMIP simulations of solar dimming, though our use of an extended thermodynamic scaling, after Byrne and O'Gorman (2015), demonstrates that local relative humidity changes play a relatively small role in the zonal mean hydrological cycle (Fig. 4). Our study reports, for the first time, that changes in tropical atmospheric dynamics dominate the P-E response to uniform solar dimming, and that these tropical rainfall shifts may be related to the interhemispheric contrast in the temperature response. This implies that the factors responsible for variations in the interhemispheric temperature response among models could ultimately explain the direction of the annual mean shift of the Intertropical Convergence Zone. This analysis is novel, and the conclusions cannot be dismissed as obvious.

In the interest of presenting a self-contained and cohesive paper, we decided to include the ensemble mean temperature change between G1 and the Preindustrial (Fig. 1), the ensemble mean P-E anomaly (G1-Preindustrial) (Fig. 2), and the ensemble mean P and E anomalies separately (Fig. 3). We now repeat our citation of Kravitz et al. 2013a where these figures appear in the manuscript to emphasize that these are not new results. These figures are included for clarity of presentation and to provide crucial context for the subsequent seven figures of novel analysis on which the paper focuses.

We greatly appreciate your comment on the importance of the seasonal cycle, as this motivated further analysis that led to additional new results. We have added the July-August-September and January-February-March streamfunction anomalies for G1-piControl (Figures 7 and 8), as well as the correlations of the ITCZ shifts with the interhemispheric temperature contrast on these seasonal timescales (panels b and c of what is now Figure 9). We have also added a new Figure 10 which shows the annual and seasonal mean positions of the ITCZ in G1 and piControl in each model. This led to the important additional conclusion that the G1 scenario damps the seasonal migration of the ITCZ by preferentially cooling the summer hemisphere and creating an anomalous Hadley circulation that transports energy to the summer hemisphere and moisture in the opposite direction. To our knowledge, this mechanism has not been discussed before or shown to exist the GeoMIP ensemble. It helps explain the finding of Tilmes et al. (2013) that summer monsoon precipitation is reduced in G1 in various regions, and presents an additional risk that needs to be considered related to implementation of solar geoengineering.

There are many comments in the attached annotated manuscript that need to be addressed.

- The typographical errors have been edited.
- One comment in the annotated manuscript asks, "Is it correct to categorize all precipitation processes this way? Is it atmospheric dynamics that changes precipitation microphysics or cloud thickness or lapse rate?" regarding our sentence: 'P-E changes not captured by this scaling are driven by non-thermodynamic mechanisms, including changes in relative humidity or atmospheric dynamics."

Decomposing the hydrological response to a climate forcing into thermodynamic and dynamic components is not the only possible way to understand the system of course, but it is a useful approach that has been fruitfully employed in other contexts (i.e., Seager et al. 2010; Wills et al. 2016; Li and Li 2014).

- Page 2, Line 29 of original manuscript: P-E over continents sets the total runoff, which includes the water flux that penetrates the surface. This has been clarified in the revised text. P-E is indeed a key factor in setting the salinity of the mixed layer and is not merely a coastal effect as you suggest, so we kept this part of the sentence intact. A citation to Byrne and O'Gorman (2015) was added here.
- Page 2, Line 30 of original manuscript: We provide the citation of the Held and Soden (2006) paper in which the P-E thermodynamic scaling is derived.
- Page 3, Line 26 of original manuscript: The comment is "Scaling is a statistical simplification. But what is physically going on? Scaling cannot predict anything. Why would you expect the physics of the situation to behave this way?"
  The P-E scaling is ultimately rooted in the Clausius-Clapeyron scaling of saturation vapor pressure with temperature. We discuss this in the beginning of Section 2.1.
- Page 3, line 28 comment: "Kravitz et al. (2013) already found this. And in the Tropics there is cooling. You need to reference previous work." We have added a citation here.
- Page 4 lines 1-2: We have added text to explain why thermodynamic scaling predicts an increase in P-E in G1-piControl. Thermodynamic scaling is based on the expectation that water vapor will increase in the atmosphere where there is warming (due to the Clausius-Clapeyron scaling of saturation vapor pressure and the assumption of constant RH), and thus that the P-E will increase in these regions. It is therefore not surprising that the thermodynamic scaling calculation results in P-E enhancement over regions with positive temperature anomalies.
- Page 4 line 8: We now define the ITCZ acronym in the Abstract.
- Page 4 line 11: The ECEARTH output files are faulty and log impossibly large precipitation values across most of the planet.
- Page 4 line 22: Edited to clarify.
- Page 7 line 5: "This is not a new finding. And why don't you show the seasonal cycle, which does not average out the interesting physics?" We now present the seasonal cycle of the Hadley circulation changes (Figs. 7, 8) as well as seasonal analysis on the ITCZ shifts (Figs. 9, 10). This paper is the first to present Hadley circulation changes in the G1 experiment.
- We have added a column with references for each model to Table 1.

In addition: I am confused. The text says there were 13 models and you excluded one, but do not say which model and why. Then Figs. 2 and 3 used 11 models, but excluded one. Again, what was the criterion for excluding the model? Table 1 only lists 12 models.

We did not have access to all of the data for CSIRO. In addition, the precipitation results for ECEARTH were not saved properly and could not be analyzed ($\delta P$ values $>>2$ mm/day over most latitudes). We now include this information in the Introduction.

Fig. 1 is not a new result. It is the same as Fig. 2 (top right) of Kravitz et al. (2013a), and this needs to be acknowledged. Figs. 2 and 3 are not new. They are the same as Fig. 5 of Kravitz et al. (2013a), and this needs to be acknowledged.

These provide background for the novel work which comprises the vast majority of the paper, including figures and discussion, as addressed above.

I don't understand which results are plotted in Fig. 4A. Is it G1? piControl? The difference? The caption just says "as simulated in the models." If it is the difference, why does it differ from the results shown in Fig. 1 of Kravitz et al. (2103a), and again not acknowledged?

Fig. 4A is a new result, and is different from Fig. 1 of Kravitz et al. (2013a) because they plot the P-E anomaly over land only (see their caption).  All four panels of Figure 4 are P-E anomalies (G1-Preindustrial).  The caption has been edited to clarify this.

Graphics are poor quality. For Figs. 1-3, 5: - The color shading has way too many shades, so it is impossible to determine the value by looking at a color on the map. Use fewer values and include labeled contour lines.

We have reduced the number of colors on the color bars in Figs. 1-3, 5 to make the results more apparent without visual strain.

- The stippling is much too dense. It is impossible to see the shading underneath it.

The stippling denotes areas where fewer than ~64% of models agree on the sign of the change, so it is not important to see the shading underneath it.  The regions with higher model agreement are intentionally emphasized.

- The x-axis label is wrong. The scaling is wrong and the longitude labels are in the wrong place. The right end should be 0∘. - The entire figure is blurry and too low resolution.

Thank you for pointing out that some of the longitude labels were misplaced.  This has been corrected.  With the changes to the color bar noted above, the figures are clear and easily readable.

- The criterion for shading varies from 62.5% to 64% to 66% agreement. Why? Why not use the 75% criterion of Kravitz et al. (2013a), which covers less of the data? - Try using GrADS. It makes beautiful maps, and automatically gives you labeling, contours, and shading.

The criterion for shading varies from 62.5% to 64% to 66% because the number of ensemble members among the figures varies based on data availability.  All necessary exclusions were noted in figure captions. The temperature ensemble mean includes 12 models, while the rest include 11 models due to the exclusion of EC-EARTH.  The relative humidity ensemble mean includes eight models.  We reached out to scientists from all the modeling groups whose data

was not available on public servers, but not everyone was able to provide the surface relative humidity.

The 75% criterion would actually cover more of the data, because it would stipple all regions for which fewer than 75% of models agree on the sign of the change, rather than the regions where fewer than ~64% of the models agree. For the purpose of conveying the scientific content, a higher resolution image will not make a worthwhile difference, as the figures are already highly legible. I will try to use GrADS for future work.

For Fig. 6, the color shading has way too many shades, so it is impossible to determine the value by looking at a color on the map. Use fewer values and include labeled contour lines. Use "piControl" rather than "Preindustrial," as it is the standard CMIP5 terminology.

We have reduced the number of colors in this figure and in the new JAS and JFM streamfunction figures in order to facilitate understanding. We now use the shorthand "piControl" in much of the paper.

**References not already included in paper**

Li, L. & Li, W. *Clim Dyn* (2015) 45: 67. doi:10.1007/s00382-014-2216-3

Seager, R., N. Naik, and G. Vecchi, 2010: Thermodynamic and Dynamic Mechanisms for Large-Scale Changes in the Hydrological Cycle in Response to Global Warming. *J. Climate,* **23**, 4651–4668, doi: 10.1175/2010JCLI3655.1.

Wills, R. C., M. P. Byrne, and T. Schneider (2016), Thermodynamic and dynamic controls on changes in the zonally anomalous hydrological cycle, *Geophys. Res. Lett*., 4640–4649, doi: 10.1002/2016GL068418.

[revised manuscript text omitted]

---

## Referee Report (RR1)

[referee-annotated manuscript omitted]

---

## Referee Report (RR2)

**Review of the revised manuscript "Thermodynamic and dynamic responses of the hydrological cycle to solar dimming" submitted for publication in ACP by Smyth et al.**

This review refers to the revised version of the manuscript as attached to the comment AC1 from Jan, 13, 2017.

The authors aim to "improve our understanding" of the effects of SRM on the hydrological cycle. To this end, they do basically two types of analyses: An estimation of effects of thermodynamics and changes in near-surface relative humidity on P-E following an approach by Byrne and O'Gorman (2015), and a simple plotting of stream functions related to the Hadley circulation. I think that there are two interesting results: a) thermodynamics have only a small influence on the P-E response patter to SRM, and b) the ITCZ shift resulting from SRM seems mainly to respond to the relative cooling of the respective summer hemisphere). In principle, I think these results are sufficiently new and interesting to warrant publications. However, I think that both presentation and analysis have a couple of deficiencies that should be dealt with before the manuscript can be considered for publication. My major issues are:

The motivation for this study in the introduction is given as "to help improve our understanding of this issue" (impact of SRM on the water cycle). I think this is much too vague. There is a large number of papers that has dealt with this issue (with respect to GeoMIP e.g. Schmidt et al., 2012, Tilmes et al., 2013, Kravitz et al. 2013; and many others with and without connection to GeoMIP), and also have stated that the model response in the tropics is less conclusive than in middle and high latitudes. I think the introduction needs to briefly summarize what the state of knowledge and what the open questions concerning "this issue" are and to provide a more specific motivation for this study.

The analysis in 2.1 mainly relies on an "extended scaling" estimate of the P-E change presented by Byrne and O'Gorman (2015). The authors just use two of the four terms of the original equation. They state that they "exclude changes in the horizontal gradient of $H_s$" but don't mention that they also exclude potential changes in the temperature gradient. It's not sufficient to argue with the "sake of simplicity", in particular when in the end the residual is interpreted as "driven by atmospheric circulation". There needs to be a discussion of why the two excluded terms are considered unimportant.

One of the main conclusions (3[rd] sentence of the "Conclusion") seems to be that "thermodynamic scaling and relative humidity changes may be important for "smaller scale responses to geoengineering". A similar statement is made at the end of section 2.2. I'm wondering why the authors do not attempt to substantiate this claim. In fact, if I haven't misread 2.2, little effort is made there to analyze the spatial pattern of the effect of relative humidity changes that goes beyond what had been said already in 2.1, although 2.2 is introduced to provide this. 2.2 tries to summarize a lot of earlier work, but it is difficult to identify a clear goal of this section and an analysis that justifies the statements

mentioned at the beginning of this paragraph. Later in the conclusions it is said that "we also present evidence that land-sea contrasts in evaporation rates, resulting in land-sea contrasts in relative humidity contribute to small changes in P-E with solar dimming". This evidence is hard to find in the manuscript. In fact, spatial patterns are in general very little discussed, as are the precipitation and evaporation patterns presented in Fig. 3. With the existing discussion of results, I don't see the use of this Figure.

The final suggestion are studies of "targeted solar geoengineering". However, my main impression is that in the perceived focus region of this study, the tropics, results seem to be very model dependent. This doesn't come as a surprise as the simulated tropical hydrological cycle is strongly influenced by parameterized convection. Earlier studies have discussed uncertainties introduced by convection schemes. I think such studies need to better referenced here. Instead of suggesting another sensitivity study that may be hampered by the same issues, I'd rather suggest to more concisely discuss potential reasons for the apparent difficulty to estimate tropical responses and suggest potential ways forward if there are any.

Minor issues:

P4l5 and l8: Why are temperature anomalies "minimal" and why mention as a contrast that hydrological effects are not eliminated? Temperature are not eliminated, either.

P4l17: It is said that the "ensemble mean reflects strong reductions … in the subtropics (Fig. 3)". I don't see such reductions in the subtropics.

P4l20 "stronger … effects that cancel out in the ensemble mean (Fig. 4A)" Stronger than what? Why not show the ensemble mean? I don't think that effects cancel out.

P5l15: This sentence is confusing. Is that true only for high vertical resolution models? Can the models of this intercomparison be considered of high vertical resolution?

P5L26 abrupt4xco2 is not a GeoMIP but a CMIP simulation.

P8l31 It is stated that the damped seasonal ITCZ migration "would likely mean a reduction of precipitation in areas …" If this is considered an important result, why not look at it in the models at hand?

P9l2 The second sentence of the conclusions is confusing because it compares two things ("thermodynamic scaling captures the general spatial structure of P-E changes under global warming and "large scale rainfall changes in … geoengineering") which seem difficult to compare. If comparing global warming and geoengineering simulations one should do the comparison with respect to the same parameters for both cases.

The caption of Fig. 4 is inaccurate in several places ("$\delta P - E$ difference" etc.).

---

## Author Response (AR3)

**acp-2016-886: Thermodynamic and dynamic responses of the hydrological cycle to solar dimming**

Reviewer Comments are written in black font, and Author Responses are in blue.

**Reviewer 1**

Thank you for your time in reviewing the article.

**Reviewer 2**

I recommend this paper be rejected. It does not present interesting new science, and there are still flaws in the figures. The analysis of differences in climate change simulations in which the global average temperature does not change, with a scaling that depends on temperature differences, does not make sense to me. I do not understand why that part is in the paper.

The decomposition of P-E into thermodynamic, relative humidity, and dynamic components helps us to compare the mechanisms underlying rainfall changes in simulations of global warming and geoengineering. While thermodynamics capture P-E anomalies in global warming simulations reasonably well, they do not explain the changes in G1. We try to explain how the dynamics and the relative humidity anomalies alter P-E in G1 relative to piControl.

I am very annoyed that the authors did not respond to one of the items in my previous review. The maps are still not plotted correctly. The longitude labels are still in the wrong place. And there is a border at the top, bottom and right edge of the maps with no shading. This gives me no confidence that the results are plotted correctly. You don't have to use GrADS, which would not have this problem, but there are many other graphics programs, such as NCL, ferret, and even Matlab that can do this. This refusal to fix this aspect of the paper alone makes me recommend to the Editor that this paper be rejected, and that if resubmitted the Editor makes sure the maps are of an acceptable quality.

In the new attached draft of the manuscript, we have corrected the longitude labels. We did not notice the white edges on the maps (due to NaN values), and have addressed this issue as well. The consistency of figures 1-3 with those in Kravitz et al. 2013a should assure readers that the data are correctly presented.

Fig. 4 has no significance measures or error bars. How different from zero would the values have to be to merit consideration?

We can measure significance based on agreement between the models because we have an ensemble. Where a majority of the models agree on the sign of the change, we consider the result robust. In Fig. 4A, most of the models show drying at 10°N and 10°S, moistening poleward of that around 30°N/ S, and drying further poleward near 50°N/ S. In Fig. 4C, some models exhibit slightly positive anomalies, and some slightly negative, so the contribution of local relative humidity changes to P-E anomalies is either negligible or very small.

p. 4, last paragraph. No, the small differences between simple and extended scaling, and the large disagreement between them and the actual results, mean that this is not an appropriate way to analyze the results. First of all, you need statistical tests to show how different the scalings need to be from each other to even deserve consideration. To say that relative humidity (RH) plays a modest role is incorrect, and certainly should not be in the abstract. What is correct is that you cannot tell how important RH is.

We specify that the role of *local changes* in RH do not play an important role, since the extended scaling accounts for the effect of local RH changes on P-E in the second term (Eq. 3). We now explicitly acknowledge in the paper that changes in the gradient of RH could be responsible for some of the G1-piControl P-E changes. This effect does not explain discrepancies in the tropical P-E response between models because the RH response is robust across the model suite.

In various places in the paper, the authors say data are not available. But did they write to the modelers to obtain the data? Just because they are not posted to the websites they looked at does not mean they do not exist. In my experience, modelers are happy to send data in response to a request.

We made an earnest effort to obtain all the data needed for this study. The Earth System Grid Federation (ESGF) server was not fully operational, so we found data on other servers, and then contacted scientists from the modeling groups to fill in the gaps. We communicated with a coordinator of GeoMIP as well. We did not make any arbitrary exclusions in our data analysis.

If the authors are going to analyze RH and ITCZ location, and their changes with geoengineering, it is incumbent on them first to analyze the piControl runs to see if the models do a good job of simulating these in the first place. If not, then how can we trust small changes. It is traditional to through out models in such a comparison if their current climate differs quite a bit from observations, not because you were not able to get the model output.

We have plotted the ITCZ position in every month in each model and in the multi-model mean (Appendix Fig. ii). This plot shows that every model exhibits a sinusoid-like seasonal migration of the ITCZ similar to that seen in observations (Waliser and Gautier, 1993, J. Clim., Figure 4h). While there are biases in the ITCZ positions in the individual models, and the annual mean position is closer to the equator in GCMs than in the observations, the overall seasonal cycle is reasonably well captured, and when this cycle is dampened in every model, we think this tells us something important about the physics of the system. For the annual mean ITCZ, our focus is not on "trusting small changes" but on examining the spread between the different models and what might be responsible for them. Biases in the representation of tropical precipitation in GCMs have been analyzed elsewhere (e.g. Stanfield et al., 2016, Clim. Dyn.). These biases have not generally been invoked to argue against the publication of papers examining changes in the ITCZ under global warming simulations, so it does not seem reasonable to require that the models perfectly represent tropical convection before we can examine any changes in them under the G1 experiment.

The ITCZ shifts found here are very small (<1°) and completely expected given the N-S temperature change differences. Since the models differ so much in their simulation of the ITCZ, this does not seem an important result.

The ITCZ changes are indeed small in magnitude. They are, however, still interesting from a scientific perspective. As we write in the paper: "Small changes in the latitudinal range and strength of the Hadley circulation and associated precipitation have large local implications, especially on subannual scales (Kang et al., 2009)." In addition, the correlation of the annual mean shifts with the N-S temperature change differences demonstrates that the paradigm of the ITCZ "shifting toward the warmed hemisphere", which has been seen in numerous other studies, also applies to solar geoengineering. Quantifying the magnitude of the ITCZ shifts in the GeoMIP ensemble should also be of interest to the broader community given the current discussions of ITCZ shifts in slab ocean vs. fully coupled models. The reduction in the seasonal migration of the ITCZ, while small in percentage terms, is robust across the different models and helps us understand how the fundamental physics of the seasonal cycle are affected by solar vs. greenhouse gas forcings.

> For the seasonal analysis, why did the authors choose the unconventional JFM and JAS or the seasons rather than the more traditional DJF and JJA? Without any special reason this was the wrong decision and prevents comparison with the results of others.

The extreme ITCZ positions occur in February and August in the multi-model mean. We include a plot in the Appendix of this Author Response to illustrate this, as well as the reduced seasonal migration of the ITCZ in G1 (Fig. ii).

> There are 15 more comments in the attached annotated manuscript that would need to be addressed.

We have addressed all of these comments.

**Reviewer 3**

1. Derivation of Eqn (2): The authors should be careful when describing the assumptions that go into this equation - the derivation assumes small meridional AND zonal gradients of CHANGES in temperature, and further assumes that the vertically-integrated atmospheric moisture scales with the near-surface specific humidity (this assumption is not currently mentioned in the manuscript). The text should be modified to state these assumptions more carefully and in full.

The assumptions implicit in equation (2) are now explained more fully in the text. Thank you for this comment.

2. Page 3, lines 27-29: The difference between simulated P-E anomalies and the extended scaling does not perfectly isolate the role of dynamics as it also includes the horizontal temperature and RH gradient terms [see the last two terms on the RHS of Eqn (7) in Byrne & O'Gorman (2015)]. These terms are particularly important over land regions, and can also be important at high latitudes over oceans (because of polar-amplified warming). This is an issue for the paper as interpreting the dynamic component of changes in P-E is difficult because it currently includes these gradient terms. I suggest the authors calculate the full extended scaling of Byrne & O'Gorman (2015) and then compare that to the simulated d(P-E) to truly isolate the dynamic component (which can

also be calculated explicitly). Calculating the gradient terms may also give insights into how large RH changes over land influence P-E.

We were unable to complete the extended scaling, which we now note in the manuscript. GeoMIP modelers will need to archive daily mean model output for this calculation to be possible in a future study.

The largest P-E changes occur in the tropics (Fig. 2). We conclude that the relative humidity changes do not explain the varying tropical rainfall responses to G1, since the relative humidity changes are robust across the model suite, while the P-E shifts are not. The Hadley circulation analysis better captures the tropical P-E responses, which is where the largest P-E anomaly occurs. We have changed the abstract to reflect this.

We now explicitly acknowledge the limitations you describe in Section 3.

3. Page 4, lines 26-27: Byrne & O'Gorman (2015) also found, in global warming simulations, that local changes in RH do not affect P-E - it might be worth connecting to that result here.

We now refer to this result from Byrne & O'Gorman (2015).

4. Page 5, lines 10-17: A recent paper focusing on land relative changes in a range of climate models might be useful for the discussion here: Byrne & O'Gorman (2016): "Understanding Decreases in Land Relative Humidity with Global Warming: Conceptual Model and GCM Simulations", J. Climate

Thank you for this suggestion. To this paragraph we have added a sentence about the Willett et al. (2014) observational study of relative humidity changes, as well as two sentences discussing the ideas from Byrne and O'Gorman (2016).

5. Page 6, lines 7-10: Calculating the impact on P-E of gradients in RH changes will allow you to check explicitly whether these land-surface affects are important for hydrological cycle changes under solar dimming conditions.

We have revised this sentence.

6. Section 2.3: It is interesting that you find narrowing tendencies for the ITCZ in these simulations. The physical processes causing changes in the width of the ITCZ under global warming have received some attention in recent years [e.g., Byrne & Schneider (2016a,b) -> see https://climatedynamics.ethz.ch/people/mike/publications.html for copies of these papers). It would be good to connect with this evolving literature when discussing the changes you see in ITCZ - would also be really cool to use the analytical framework of Byrne & Schneider to diagnose what processes in these solar dimming simulations are driving the ITCZ width changes! This is beyond the scope of the current article but could be an interesting avenue to explore in the future.

We now refer to the two suggested papers in the last paragraph of Section 2.3.

7. Page 9, lines 14-16: I think it is difficult to be confident in this statement without calculating how gradients of changes in RH affect P-E over land - they may be an important influence compared to just the local RH changes [as is the case in Byrne & O'Gorman (2015)].

We have revised this paragraph.

8. Page 8, lines 27-29: A damped seasonal ITCZ migration is one possible explanation for reduced monsoon precipitation but there are several others (e.g., reduced monsoon circulation strength, zonal shifts in the monsoon, reduced moisture content). Without further analysis I think the authors should remove this statement (and a similar statement in the Conclusions).

On page 8, we now acknowledge that the reduced ITCZ migration possibly contributes to (but does not entirely explain) the results reported by Tilmes et al. 2013, and have removed the statement from the Conclusions.

**Reviewer 4**

Major Issues:

1. "The motivation for this study in the introduction is given as "to help improve our understanding of this issue" (impact of SRM on the water cycle). I think this is much too vague. There is a large number of papers that has dealt with this issue (with respect to GeoMIP e.g. Schmidt et al., 2012, Tilmes et al., 2013, Kravitz et al. 2013; and many others with and without connection to GeoMIP), and also have stated that the model response in the tropics is less conclusive than in middle and high latitudes. I think the introduction needs to briefly summarize what the state of knowledge and what the open questions concerning "this issue" are and to provide a more specific motivation for this study."

We thank the reviewer for this suggestion, which has improved the framing of the paper. We have added several background paragraphs to the end of the introduction to describe the relevant results presented in Kleidon et al. 2014, Kravitz et al. 2013b, Tilmes et al. 2013, Bala et al. 2008, and Schmidt et al. 2012.

2. The analysis in 2.1 mainly relies on an "extended scaling" estimate of the P-E change presented by Byrne and O'Gorman (2015). The authors just use two of the four terms of the original equation. They state that they "exclude changes in the horizontal gradient of Hs" but don't mention that they also exclude potential changes in the temperature gradient. It's not sufficient to argue with the "sake of simplicity", in particular when in the end the residual is interpreted as "driven by atmospheric circulation". There needs to be a discussion of why the two excluded terms are considered unimportant.

We now use more accurate language in this section. It would certainly be valuable to calculate the full scaling to quantify the role of relative humidity more fully, but it is not possible with the

existing GeoMIP simulations, as daily mean output was not archived for most models in G1. However, the combination of our extended scaling with the analysis of the spatial distribution of relative humidity anomalies (Section 2.2) allows us to identify the regions where the gradients might play an important role. We then focus on understanding the tropical variability amongst the models, which is well explained by our analysis of the Hadley circulation and ITCZ shifts. Where relative humidity changes are large, they are also robust, so we know that the inter-model spread is rooted in dynamics.

3. "One of the main conclusions (3rd sentence of the "Conclusion") seems to be that "thermodynamic scaling and relative humidity changes may be important for "smaller scale responses to geoengineering". A similar statement is made at the end of section 2.2. I'm wondering why the authors do not attempt to substantiate this claim. In fact, if I haven't misread 2.2, little effort is made there to analyze the spatial pattern of the effect of relative humidity changes that goes beyond what had been said already in 2.1, although 2.2 is introduced to provide this. 2.2 tries to summarize a lot of earlier work, but it is difficult to identify a clear goal of this section and an analysis that justifies the statements mentioned at the beginning of this paragraph. Later in the conclusions it is said that "we also present evidence that land-sea contrasts in evaporation rates, resulting in land-sea contrasts in relative humidity contribute to small changes in P-E with solar dimming". This evidence is hard to find in the manuscript. In fact, spatial patterns are in general very little discussed, as are the precipitation and evaporation patterns presented in Fig. 3. With the existing discussion of results, I don't see the use of this Figure."

Thank you for this comment. We have reorganized the paper so that the extended scaling, which quantifies the impact of local relative humidity changes on P-E, is in Section 2.2. In addition, we have added text to Section 2.2 where we describe the reductions in relative humidity over South America and sub-Saharan Africa. We also elaborate on the role of relative humidity in paragraph 3 of the conclusions. Given the small deviations of the extended scaling from the simple scaling, and that the signs of the deviations vary amongst models (Fig. 4C), we now more carefully state that local relative humidity changes *may* play a small role in the P-E response to G1.

4. The final suggestion are studies of "targeted solar geoengineering". However, my main impression is that in the perceived focus region of this study, the tropics, results seem to be very model dependent. This doesn't come as a surprise as the simulated tropical hydrological cycle is strongly influenced by parameterized convection. Earlier studies have discussed uncertainties introduced by convection schemes. I think such studies need to better referenced here. Instead of suggesting another sensitivity study that may be hampered by the same issues, I'd rather suggest to more concisely discuss potential reasons for the apparent difficulty to estimate tropical responses and suggest potential ways forward if there are any.

We have now added references regarding the sensitivity of tropical convection to the convection scheme, and suggest in a new sentence at the end that improvements in model representation of convection and clouds could help improve our understanding of hydrological cycle changes under solar geoengineering. However, we have not taken out the suggestion of additional sensitivity studies because we think these can be useful even if there are inter-model differences.

Minor Issues

P4l5 and l8: Why are temperature anomalies "minimal" and why mention as a contrast that hydrological effects are not eliminated? Temperature are not eliminated, either.

Thank you for this question. We have changed the phrasing in several sentences in these paragraphs to emphasize that neither temperature nor P-E are entirely restored to preindustrial values. However, temperature anomalies are smaller than P-E changes, relative to their climatological values. The G1 experiment is designed to minimize temperature anomalies due to elevated carbon dioxide concentrations. In the tropics, where temperature anomalies are smallest (less than 1 K), the P-E anomalies are largest.

P4l17: It is said that the "ensemble mean reflects strong reductions … in the subtropics (Fig. 3)". I don't see such reductions in the subtropics.

This sentence has been made more specific: "The ensemble mean precipitation response reflects strong reductions in subtropical precipitation across the Pacific Ocean (Fig. 3)." The red color around 15° N/S across the Pacific Ocean denotes drying, which is consistent with a narrowing of the ensemble mean ITCZ.

P4l20 "stronger … effects that cancel out in the ensemble mean (Fig. 4A)" Stronger than what? Why not show the ensemble mean? I don't think that effects cancel out.

We have changed the phrasing in this paragraph to clarify the meaning. We mean that since zonal mean P-E shifts northward in some models and southward in others, the amplitude of the ensemble mean P-E anomalies is reduced relative to those individual model results. Figure 2 shows the ensemble mean P-E spatial pattern, and Figure 4A shows the zonal mean for individual models. The ensemble zonal mean is included for reference in the Appendix of this Author Response (Fig. i). The pattern over the Pacific Ocean dominates the zonal mean picture, and the amplitude of the changes is within 0.2 mm/day. Many of the individual models in Fig. 4A have much stronger zonal mean responses (approaching 0.6 mm/day).

P5l15: This sentence is confusing. Is that true only for high vertical resolution models? Can the models of this intercomparison be considered of high vertical resolution?

This sentence has been eliminated in favor of the three new sentences describing relative humidity analysis by Willett et al. (2014) and Byrne and O'Gorman (2016).

P5L26 abrupt4xco2 is not a GeoMIP but a CMIP simulation.

"GeoMIP" has been changed to "CMIP5."

P8l31 It is stated that the damped seasonal ITCZ migration "would likely mean a reduction of precipitation in areas …" If this is considered an important result, why not look at it in the models at hand?

While we discuss reduced seasonal migration as a possible explanation for the overall narrowing of tropical precipitation, the effect may be too small to show up strongly in specific land areas.

A detailed regional analysis is beyond the scope of this paper, which we now mention in the Conclusions.

P9l2 The second sentence of the conclusions is confusing because it compares two things ("thermodynamic scaling captures the general spatial structure of P-E changes under global warming and "large scale rainfall changes in … geoengineering") which seem difficult to compare. If comparing global warming and geoengineering simulations one should do the comparison with respect to the same parameters for both cases.

This was a problem of poor phrasing; we indeed mean to compare P-E changes in the two climate states.  We have changed the sentence to read: "While thermodynamic scaling captures the general spatial structure of P-E changes under global warming, it does not do so for idealized simulations of solar geoengineering."

The caption of Fig. 4 is inaccurate in several places ("dP – E difference" etc.).

The caption of Fig. 4 means to say both "delta" and "difference" because it represents the difference of the P-E anomaly for G1-piControl calculated by the extended scaling, minus the G1-piControl P-E anomaly predicted by the simple scaling.  We have modified the caption to hopefully enhance clarity.

**Appendix**

[Figure]

Fig. i. The ensemble zonal mean P-E anomaly
for G1 minus piControl.

[Figure]

Fig. ii. The monthly ITCZ location in individual models and the
ensemble mean (bold lines) for both G1 and piControl.

[revised manuscript text omitted]